# Type I interferon drives T cell responses to amyloid beta in the central nervous system

Julius J. Michel[1,2,10], Khwab Sanghvi [1,2,3,10], Jakob Rosenbauer[1,2], Lea Humbs[1,3], Clara Tejido Dierssen[1,2,3], Saskia Grudzenski-Theis[2], Viktoria Sachs[1,2], Kristine Jähne[1,2], Karoline Degenhardt [4], Lutz Frölich [5], Jochen Herms [6], Marc Fatar[2], Michael Platten [1,2,7,8,9] & Lukas Bunse [1,2,9] ✉

Amyloid beta (Aβ) plaque deposition in the central nervous system (CNS) is a hallmark of Alzheimer's disease (AD) and cerebral amyloid angiopathy (CAA), triggering robust innate immune responses. However, the role of the adaptive immune system remains less well understood. Here we show the immune microenvironment dynamics in APP23 transgenic (APP23-tg) mice modelling CNS amyloid pathology, using single-cell transcriptomics. We observed a marked increase in T-cell populations during late disease stages, particularly CD8+ T-cells that clustered around Aβ plaques, suggesting a targeted immune response. Among these, we identified an Aβ plaque-associated subset of CD8+ T cells expressing interferon-stimulated genes (ISGs), which promoted Type-I interferon signaling. This subset also produced CXCL10, facilitating the recruitment of non-ISG T cells through the CXCL10-CXCR3 axis. Importantly, similar Type-I interferon responses were detected near plaques in human CNS amyloid pathology. Together, these findings highlight a shift from microglia-driven to T-cell-mediated neuroinflammation as amyloid pathology progresses, with implications for time-resolved therapy development.

The abnormal deposition of Aβ as extracellular parenchymal plaques in the CNS or vascular amyloid serves as a pathological hallmark in Alzheimer's disease (AD) and cerebral amyloid angiopathy (CAA), respectively, ultimately resulting in neurodegeneration and dementia. The observation that many AD risk genes are linked to the innate immune functions of microglia, coupled with the identification of elevated levels of inflammatory markers in the brains of individuals with AD suggests that neuroinflammation plays a pivotal role in the development of neurodegenerative diseases related to Aβ[1–5]. Microglia are the resident macrophages of the brain and act as the first line of defence against CNS insults. Both, in humans and mice, it has been

shown that through pattern-recognition receptors (PRRs), microglia can recognize and subsequently surround phagocyte Aβ species[6–8]. Upon Aβ-uptake microglia elicit distinct gene expression profiles characterized by increased expression of genes linked to phagocytosis and lipid metabolism, differentiating them from homeostatic brain microglia. Several such disease-associated phenotypic states have been previously described[9–11]. However, as the disease progresses, chronic activation and excessive accumulation of Aβ drive microglia into a pathological state in which their ability to uptake Aβ is compromised[12]. A subset of microglia undergoes a phenotypic shift marked by a change in their gene expression profile towards an

[1]Clinical Cooperation Unit Neuroimmunology and Brain Tumor Immunology, German Cancer Research Center (DKFZ), Heidelberg, Germany. [2]Department of Neurology, MCTN, Medical Faculty Mannheim, Heidelberg University, Mannheim, Germany. [3]Faculty of Biosciences, Heidelberg University, Heidelberg, Germany. [4]Department of Neurology, University Medical Center Hamburg-Eppendorf, Hamburg, Germany. [5]Division of Geriatric Psychiatry, Central Institute of Mental Health, University of Heidelberg, Mannheim, Germany. [6]Center for Neuropathology and Prion Research, Ludwig-Maximilians University Munich, Munich, Germany. [7]Immune Monitoring Unit, National Center for Tumor Diseases (NCT), Heidelberg, Germany. [8]Helmholtz Institute of Translational Oncology (HI-TRON), Mainz, Germany. [9]DKFZ Hector Cancer Institute at the University Medical Center Mannheim, Mannheim, Germany. [10]These authors contributed equally: Julius J. Michel, Khwab Sanghvi. ✉e-mail: l.bunse@dkfz-heidelberg.de

enhanced expression of Type I interferon (IFN)-stimulated genes (ISG)[13]. It has been demonstrated that this population of microglia enhances neuroinflammation and contributes to synapse loss in mouse models of AD, which could be rescued by blocking the Type-I-IFN response[11]. Moreover, the same population has also been observed in human AD samples[14]. Type-I IFN signalling in microglia, therefore, seems to represent a key feature of amyloid-related neuroinflammation[15].

In contrast to microglia, the role of the adaptive immune response elicited by Aβ is much less understood. For years, the brain had been seen as an immunologically privileged organ, separated from the immune system in the periphery of the human body. Emerging evidence of a dysfunctional blood-brain-barrier (BBB), and cues of vascular inflammation in AD and other neurodegenerative diseases, have fueled the hypothesis that, besides brain resident microglia, additional blood-derived immune cells are taking on a pivotal role in the choreography of neuroinflammation[16–19]. The observations of substantial T-cell infiltration into the CNS parenchyma and increased levels of T-cells in the leptomeninges and the cerebrospinal fluid (CSF), in both mice and humans, underscore the substantial involvement of the adaptive immune system, particularly T-cells[20–28]. This immune response appears to be driven by CD8+ T-cells, which predominantly infiltrate limbic structures, in particular the hippocampus[23,25,28]. Delineating whether T-cells play a beneficial or detrimental role in AD, however, remains an open question in the field. On one hand, a study showed an acceleration of amyloid deposition, alteration in microglial phenotype, and enhanced neuroinflammation in a T-cell deficient mouse model of AD, which could be reversed upon bone marrow transplantation[29]. On the other, reports have observed a positive correlation between the number of infiltrating T-cells and the extent of neuronal loss, and that depletion of pan T-cells halted neuronal loss in mouse models[28]. Additionally, the number of a subset of antigen-experienced CD8+ T memory cells with a cytotoxic and proinflammatory phenotype − T-effector memory CD45RA+ (TEMRA) − in the CSF correlated with lower cognition scores in AD patients[25].

In addition, it remains unknown which signalling cues recruit T-cells, how T-cell effector functions engender damage or protection, and what fate awaits T-cells in the CNS. Deciphering these questions in a disease characterized by two main abnormalities − Aβ plaques and neurofibrillary tangles, which each trigger different microenvironmental reactions, is complex. It is, therefore, of relevance to deconvolute how T-cells behave in response to individual pathological hallmarks to unravel mechanisms that may better educate therapeutic interventions targeting neuroinflammatory fallout. In this study, we investigate longitudinal changes in the brain immune cell landscape along the amyloid disease trajectory with a focus on local T-cell phenotype adaption and find an Aβ plaque-associated subset of CD8+ T-cells, driving Type-I interferon responses locally. These T-cells contribute to a shift from microglia-driven to T-cell-mediated neuroinflammation as amyloid pathology progresses.

## Results

### Progressive amyloid disease drives T-cell accumulation in APP23-tg mice in a disease-dependent manner

Several murine models have been developed to mimic the pathological deficits and microenvironment alterations of AD[30,31]. With an interest in delineating neuroinflammatory responses to Aβ, we utilized a model where the leading neurodegenerative trigger was CNS β-amyloidosis. The APP23-tg mouse model is widely recognized for its relevance to not just AD but also CAA and has been extensively utilized to emulate and study the impact of Aβ deposition[32,33]. This model carries the pathogenic Swedish mutation, which increases the cleavage of amyloid precursor protein (APP) by beta-site APP cleaving enzyme-1 (BACE1), leading to enhanced production of Aβ peptides. Over time, these peptides accumulate, resulting in the formation of disease-

characteristic amyloid plaques both in the brain parenchyma and CNS vasculature[32]. To understand the temporal dynamics of β-amyloidosis, we longitudinally characterized Aβ plaque burden. Within our mouse colony, APP23-tg mice first start to form Aβ plaques by the age of 8–12 months, marking the onset of disease (Fig. S1a). Amyloid deposition was observed as senile plaques within the brain parenchyma, referred to as "parenchymal plaques", as well as in blood vessels, referred to as "vascular amyloid deposits" (Fig. S1d). A substantial, progressive increase in plaque accumulation occurred in the brain parenchyma as well as in the number of amyloid-affected blood vessels over time. Female animals typically displayed higher parenchymal plaque burdens compared to males, but this difference was not significant in the number of affected blood vessels (Fig. S1a, b). Plaque deposition overwhelmingly occurred in the brain cortex with a slight decline as the disease advanced, whereas was augmented in regions associated with subcortical memory systems, such as the striatum and hippocampus, over time (Fig. S1c). APP23-tg mice have been shown to demonstrate age-dependent alterations using the Morris Water Maze test[34,35]. We additionally performed Nestlet assays to examine neurological deficits and showed a decrease in nesting scores with age. Nesting ability was significantly hindered in APP23-tg mice compared to wildtype at the age of 24 months after 24 h of nestlet placement and across most time points after 7 days of nestlet placement (Fig. S1e, f, h). This deficit was more pronounced in female animals with lower scores across all time points 24 h after nestlet placement, and for 20- and 24-month-old mice after 7 days of nestlet placement (Fig. S1g, i). Thus, APP23 mice provide a model that resembles progressive amyloidosis and associated age-dependent neurological alterations. Furthermore, this model does not exhibit the formation of neurofibrillary tangles, ensuring that our subsequent findings are primarily due to inflammatory responses triggered by Aβ plaques.

Based on our assessment of Aβ plaque deposition dynamics in APP23-tg mice, we determined mice aged 12–14 m and 20–24 m as having "early" and "late" amyloid disease, respectively. This nomenclature has been used subsequently. Next, to understand the changes in the immune microenvironment with progressive disease, we subjected CD45+ cells, isolated from the brains of early and late mice and corresponding wild-type littermates (Wt), to single-cell RNA (scRNA) and VDJ (scVDJ) sequencing (Figs. 1a and S2a). The resulting 20 transcriptionally distinct cell clusters, obtained from a total of 21,156 cells, were annotated based on expression of canonical markers for each immune cell phenotype (Figs. 1b and S2b, c). Within the microglial subsets, we identified disease-associated microglia (DAM) and Type-I interferon-stimulated gene signature microglia (ISG MG) − two microglia subtypes that have been previously described in AD models[9,11]. Interestingly, we observed a significant increase in the abundance of T-cells in late disease, markedly CD8+ T-cells (Fig. 1c) which was also reflected in the majority of the CD8+ T-cell subclusters, both in terms of frequency of CD45+ cells (Figs. 1d and S3) as well as absolute numbers (Fig. 1e). The CD4+ T-cell response was mostly driven by T-helper-17 cells composed within the IL17+ T-cell cluster. Underscored by a strong expression of interleukin-17 (Il17)-a, Il17f, and Rorc, the cluster also contained a small fraction of gamma-delta T-cell receptor T-cells (Fig. S4). The observed increased abundance of IL17+ T-cells in late disease goes in line with previous reports that have elucidated the pathological neuroinflammatory role of IL17-signaling in AD and other neurodegenerative disorders[36–41]. Ingenuity pathway analysis (IPA) of the single-cell RNA sequencing (scRNAseq) data further showed that pathways upregulated in early disease were associated with the development, migration and effector responses of myeloid cells whereas late disease was highlighted by migration, activation and effector function of lymphocytes (Fig. 1f). These findings underline an overall shift from a microglia-driven to a T-cell-mediated neuroinflammatory response with progressive amyloid disease.

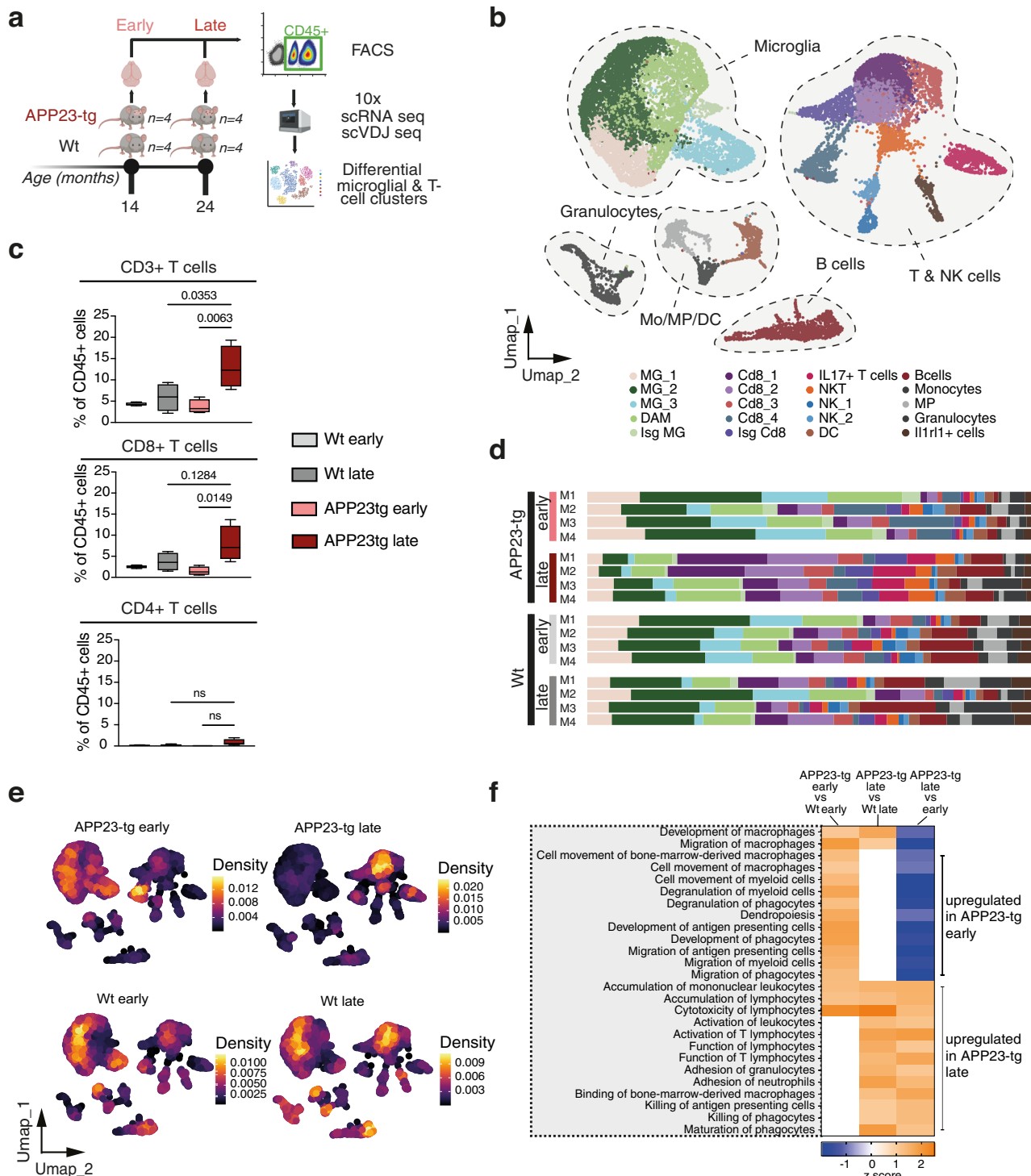

**Fig. 1 | T-cell accumulation increases with Aβ plaque burden in APP23-tg mice.**
**a** Schematic of single-cell RNA sequencing (scRNAseq) workflow. Brain parenchymal CD45+ immune cells were isolated from APP23 transgenic (APP23-tg, *n* = 8) and age- and sex-matched wild-type (WT, *n* = 8) littermates. Mice were stratified into four experimental cohorts (two timepoints × two genotypes) with four biological replicates per group (Created in BioRender. D170, P. (2026) https://BioRender.com/s45bl6s). **b** Uniform manifold approximation and projection (UMAP) plot of CD45+ immune cells from brain parenchyma of all mice. **c** Quantification of CD3+ T cells, CD4+ T cells, and CD8+ T cells as a percentage of total CD45+ cells. Box plots show median with boxes indicating 25th–75th percentile IQR and whiskers indicating min to max values. Statistical analysis was

performed using two-way ANOVA followed by Tukey's multiple comparisons test. **d** Bar plot showing the relative abundance of each identified immune cell type, expressed as percentage of total CD45+ cells per individual mouse (M1-M4). **e** Uniform manifold approximation and projection (UMAP) plot displaying a density gradient representing absolute immune cell counts across all four conditions. Individual animals are not shown separately. **f** Ingenuity Pathway Analysis (IPA) of CD45+ immune cells comparing early-stage APP23-tg vs WT, late-stage APP23-tg vs WT, and early vs late-stage APP23-tg mice. Differentially expressed pathways (z-score > 1.5 or <−1.5) were identified using IPA's "Diseases and Functions" module. Color scale indicates predicted pathway activity: orange for upregulation, blue for downregulation.

Importantly, this age-dependent shift was independently recapitulated in our targeted spatial transcriptomics dataset. Stratifying immune cell composition by proximity to Aβ plaques revealed a predominantly microglia-associated response in plaque-adjacent regions at early stages, which transitioned toward increased T-cell representation in late disease (Fig. S6f, g).

## T-cell accumulation is clustered around Aβ plaques

We were curious whether the T-cell response was directly associated with Aβ. To assess whether the T-cell response was directly associated with Aβ deposition, we probed T-cell infiltration longitudinally by histologically staining for Aβ, T-cells (CD3), and endothelial cells (CD31) to differentiate between parenchymal and vascular amyloid deposits (Figs. 2a and S5). We observed a gradual increase in T-cell abundance with age, corresponding to increasing plaque burden and in concurrence with our transcriptomics data, peaked at 20 m (Fig. 2b, c). This increase was coupled with an increase in the number of T-cells per plaque, with a significant increase in parenchymal plaques compared to vascular amyloid deposits (Fig. 2d). These infiltrating T-cells were found in close proximity to, and clustered around Aβ plaques (Fig. 2e), but were absent in plaque-devoid regions (Fig. 2f). The number of T-cells quantified per histological section significantly correlated with plaque burden, with a preferential association with parenchymal plaques compared to vascular amyloid deposits (Fig. 2g). This data suggests that although the T-cell response is directed towards Aβ, infiltrating T-cells have a greater attraction towards parenchymal plaques compared to vascular amyloid deposits. As vascular amyloid deposits are, in principle, more accessible for blood-born T-cells, this phenomenon could be attributed to the greater size of parenchymal Aβ deposition, the distinct composition of the plaques with $A\beta_{42}$ predominantly found in parenchymal plaques, $A\beta_{40}$ predominantly found in vascular amyloid deposits, or specific features in the surrounding immune microenvironment.

## ISG CD8+ T-cell population defines the late-disease Type-I interferon response

Type-I interferons are well-known drivers of autoimmune inflammation (Decout et al., 2021). In the context of anti-amyloid response, ISG microglia have been described as the harbingers of the interferon response and were also discernible in our dataset (Fig. 3a, b). However, by differential gene expression (DGE) analyses, we found a T-cell subset in our scRNAseq dataset with elevated ISG expression. This subset upregulated known canonical ISGs compared to other T-cells, similar to ISG microglia (Fig. 3c). Contrastingly, however, where ISG microglia were more abundant in early disease, ISG CD8+ T-cells were prominent in late disease (Fig. 3b, d). We then looked at the distribution of the expression of a large set of ISGs between all transcriptomic clusters in an unbiased manner. The ISGs grouped into 2 sets, with a subset of genes enriched in ISG microglia and the other subset enriched in ISG CD8 T-cells (Fig. 3e). The expression of ISGs in other clusters was not profound. A few ISGs, such as Isg15, Ifit2, and Bst3, overlapped between microglia and T-cells. We curated the genes differentially expressed in ISG CD8+ T-cells as an ISG T-cell specific signature and found it significantly augmented in late disease compared to early as well as Wt brain immune cell infiltrates (Fig. 3f). Altogether, this contrast underscores a shift in the arbiters of neuroinflammatory type-I interferon response along the progression of amyloid disease.

We next utilized targeted single-cell spatial transcriptomics (Molecular Cartography) to evaluate the localization of ISG expression in amyloid-burdened mouse brains (Supplementary Table 1). A plaque neighborhood analysis was performed, by first designating Aβ plaque positions and then determining regions of interest (ROIs) based on 3 distance metrics—"inside" (within the plaque and on the boundary), "adjacent" (up to a distance of 69 μm from the plaque boundary corresponding to 500px on rendered images of assay slides) and

"outside" (beyond 69 μm) in a cell-segmentation free manner. ISG was strongly induced in the plaque neighborhoods (Fig. 3g), where upregulation was observed in a subset of plaque neighborhoods in early disease; the majority of plaque neighborhoods in late disease had significantly upregulated ISG expression (Fig. 3h). To delineate the relative contribution of microglia and T-cells to ISG expression, we quantified ISG expression levels as a measure of distance from microglia and T-cell markers, respectively, within the plaque neighborhoods and ascertained a stronger activation of ISGs in T-cells (Fig. 3i, j). Similarly, cell-segmentation-based analyses confirmed an ISG-upregulating T-cell subset. When compared to other cells, the abundance of cells within this subset was higher in the plaque neighborhood (<69 μm) (Fig. S6a–j). Overall, these observations identify ISG CD8+ T-cells as a T-cell subset driving Type-I interferon response to Aβ plaques in late disease.

Analyses using the Cell Chat algorithm revealed a heightened interaction between microglia and CD8+ T-cells in early disease brains compared to cognate Wt brains as expected, markedly from ISG microglia and DAM (Fig. 3k). The global increase in both the number and strength of interactions in late disease as well as inflated participation of T-cell subsets alluded to a strong adaptive immune system-driven neuroinflammatory local reaction upon progressive amyloidosis (Fig. 3l).

## ISG CD8+ T-cell derived CXCL10 steers T-cell trafficking via the CXCL10-CXCR3 axis

We next assessed whether the induction of ISGs in a subset of CD8+ T-cells was a bystander phenomenon of interaction between T-cells and ISG microglia or whether it had an additional functional relevance. CXCR3 is a chemokine receptor highly expressed on effector CD8+ T-cells and regulates their function and trafficking into sites of inflammation[42]. A previous report had identified CD8+ T-cells infiltrating into brains, highly expressed CXCR3[19]. CXCR3 activation is driven by chemokine ligands – CXCL9, CXCL10, and CXCL11. Interestingly, DGE analysis of ISG cells compared to other cells had highlighted upregulated CXCL10 expression (Fig. 3a, c), prompting us to hypothesize that this chemokine axis was responsible for T-cell infiltration in response to Aβ. Deconvoluting interactions between ISG CD8 T-cells and other microglial and T-cell subsets, we observed chemokine-chemokine receptor interactions upregulated, among others, e.g., Cxcl10-Cxcr3, Ccl5-Ccr3, Ccl4-Ccr3 (Fig. 4a). Since Cxcl10-Cxcr3 interactions were upregulated, we investigated the network analysis of interactions within this domain in late disease between all cell subsets. ISG CD8 T-cells were one of the major drivers, targeting other T-cell populations (Fig. 4f). On mapping Cxcr3 expression among cell subsets, the highest expression was observed in CD8_1, 2, and 3 and NKT cells, populations which also were more abundant in late disease (Fig. 4b, c). Conversely, Cxcl10 was primarily expressed by ISG CD8+ T-cells (Fig. 4b, c). At a spatial resolution, Cxcl10 expression was dense around the plaques (Fig. 4d) and significantly enriched in the plaque neighborhood (Fig. 4e). When Cxcl10 transcript abundance was measured as a metric of distance from ISG transcripts, we found co-expression of both within the same cells in late disease (Fig. 4g), firmly in line with the dynamics of ISG CD8 T-cell residence and localization. To check if CXCL10 was also produced by other cell types, such as astrocytes in the brain, we checked expression levels post-cell-segmentation and found CXCL10 sourced solely from ISG T-cells (Fig. S6g). These observations identify ISG CD8+ T cells as a dominant CXCL10-producing population in progressive amyloid disease.

To functionally assess whether CXCL10-CXCR3 signaling was required for T-cell chemotaxis, we performed retroviral CRISPR-Cas9-mediated CXCR3 knockout in primary splenic T cells from Cas9-expressing mice, achieving ~40% transduction efficiency (BFP+; Fig. S9a). Flow cytometric profiling confirmed robust loss of CXCR3 surface expression in CXCR3 gRNA-transduced cells compared

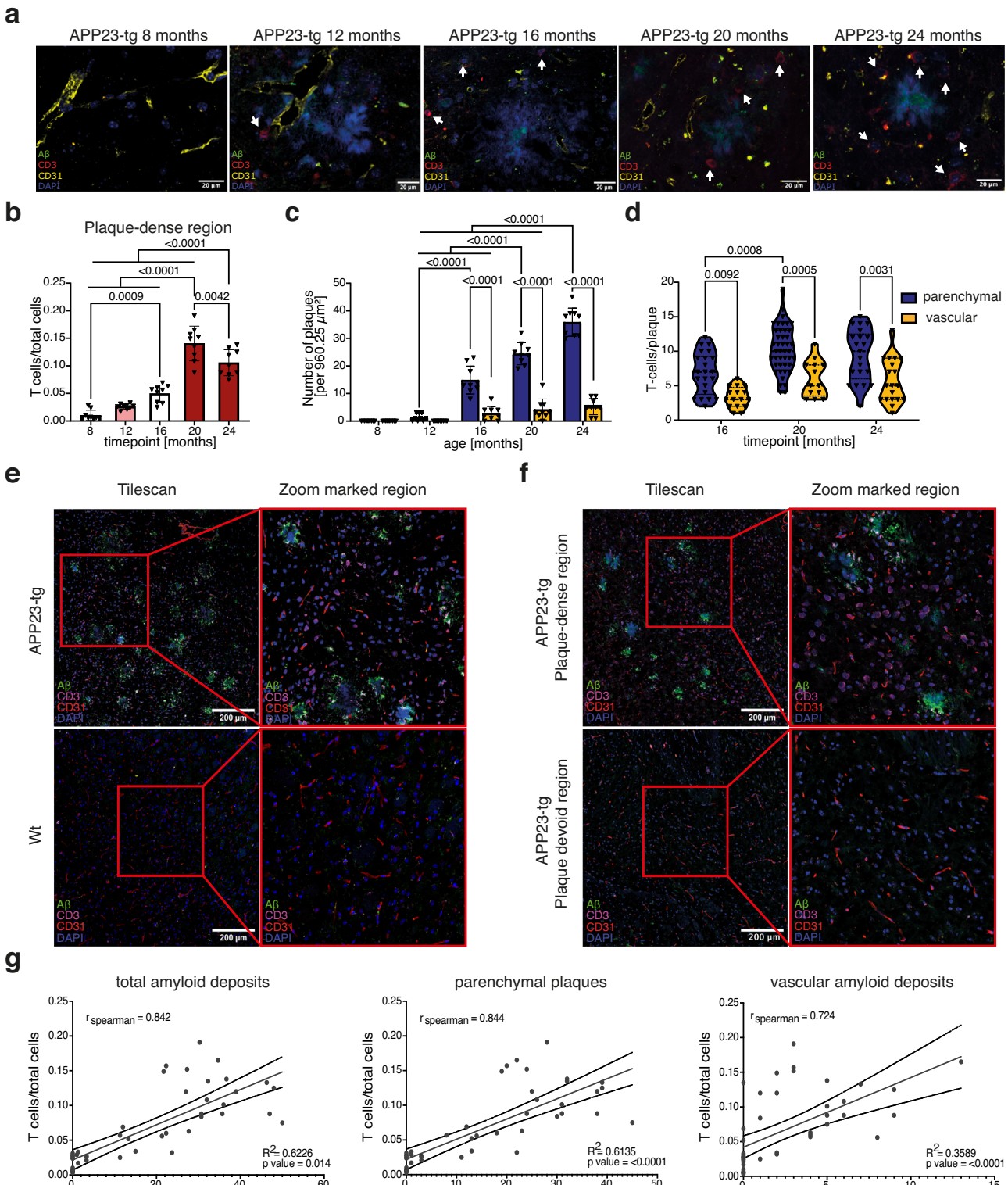

**a**

APP23-tg 8 months | APP23-tg 12 months | APP23-tg 16 months | APP23-tg 20 months | APP23-tg 24 months

**g**

total amyloid deposits

$r_{spearman} = 0.842$

$R^2 = 0.6226$
p value = 0.014

parenchymal plaques

$r_{spearman} = 0.844$

$R^2 = 0.6135$
p value = <0.0001

vascular amyloid deposits

$r_{spearman} = 0.724$

$R^2 = 0.3589$
p value = <0.0001

with non-targeting (NT) controls (Fig. S9b, c). In transwell migration assays, addition of recombinant CXCL10 to the basolateral chamber elicited strong migration of NT cells, whereas CXCR3 knockout markedly reduced migration (Fig. 4h). CXCR3-blocking antibody treatment in NT cells similarly diminished chemotaxis, demonstrating CXCR3-dependent responsiveness to CXCL10 (Figs. 4h and S9d). No migration occurred in medium-only controls. Because ISG CD8+ T cells were a major source of CXCL10 in late amyloid disease in vivo, we next tested the chemotactic potential of ISG CD8+ T-cells. Murine CD8+ T-cells were placed in the basolateral chamber and stimulated with the STING agonist DMXAA for 5h to induce robust ISG activation[43,44]. ISG-induced CD8+ T-cells drove strong transmigration of NT CD8+ T cells, whereas transmigration of CXCR3 KO cells was markedly reduced (Fig. 4i). Untreated cells used as control drove minimal transmigration. These data show that ISG-activated CD8+ T cells possess intrinsic chemotactic activity, at least partially dependent on CXCR3. Together, our in vitro results demonstrate that both exogenous CXCL10 and ISG-induced CD8+ T-cells function as potent chemotactic stimuli, and that CXCR3 expression is important for efficient T-cell migration. These findings provide direct functional support for our in vivo observations

**Fig. 2 | Brain infiltrating T-cells localize to Aβ plaque neighborhoods.**
**a** Representative confocal images of fresh frozen paraffin-embedded (FFPE) cortical sections from APP23-tg mice aged 8, 12, 16, 20, and 24 months. Regions of interest are shown at 20× magnification (LSM700 confocal microscope). T cells (CD3+) are indicated by arrows and identified based on marker expression and characteristic morphology. The strong signal observed in the DAPI channel at plaque sites reflects the known intrinsic blue fluorescence of plaque cores[64]. **b** Quantification of CD3+ T cells normalized to total cell count across five time-points (8, 12, 16, 20, and 24 months). Statistical analysis was performed using ordinary one-way ANOVA with Tukey's multiple comparisons test. **c** Quantification of parenchymal and vascular amyloid deposits per tile scan across the same five timepoints. Two-way ANOVA with Tukey's post hoc test was used for statistical analysis. For 2 (b) + (c) data are presented as mean ± SD. **d** T-cell count per amyloid plaque (parenchymal vs vascular) in APP23-tg mice aged 16, 20, and 24 months. Statistical significance was assessed using two-way ANOVA with Tukey's post hoc

test. For analyses in (**b**–**d**), three tile scans per section from three sections per mouse ($n = 3$ mice per timepoint) were used. Each tile scan covered an area of 960.25 μm². **e** Fixed frozen immunofluorescence (IF) images of cortical sections from one representative 24-month-old APP23-tg mouse (top) and age-matched wild-type control (bottom). Left: tile scan at 20× magnification; right: zoomed-in area of interest. **f** Fixed frozen IF images of a 24-month-old APP23-tg mouse showing a plaque-rich region (top) and a plaque-devoid region (bottom). Left: tile scan at 20× magnification; right: corresponding zoom-in. For 2 (e) + (f) three tile scans per section from three sections per mouse ($n = 3$ mice per timepoint) were used. **g** Spearman correlation analysis between T-cell frequency and amyloid plaque burden, including parenchymal (middle), vascular (right), and total plaque counts (left). The solid line shows the linear regression fit (least-squares), with curved lines indicating the 95% confidence interval of the fit. Spearman's $\rho$, $p$-value, and $R^2$ are indicated in each plot.

and indicate that the CXCL10-CXCR3 axis is a mechanistically significant driver of T-cell infiltration in late-stage amyloid neuroinflammation, while not excluding additional pathways.

## Activation-induced exhaustion characterizes CD8+ T-cells in the plaque neighborhood

Our IPA analysis, as well as Cell Chat analysis on top regulated pathways, revealed that late disease was characterized not only by infiltration of CD8+ T-cells but also by increased activation, proliferation, and cytotoxicity (Figs. 1f and 5a). Markers of T-cell activation and effector functions such as Nkg7, Gzmk, and Ccl3 were strongly upregulated in late disease, associated with the Cd8_1 T-cell subcluster (Fig. 5b, c) and expressed in propinquity to plaques (Fig. 5d). Interestingly, Cd8_3 exhibited a different pattern of activation highlighted by Gzma, Gzmb and Prf1, which were also upregulated in Wt brains (Fig. 5b). Additionally, the abundance of this subcluster did not differ between late disease and Wt, converse to Cd8_1 which was significantly more abundant in late disease (Fig. S3). This is suggestive of Nkg7 and Gzmk expression as highlighters of trafficking T-cells in cue to CXCL10-induced chemotaxis in response to amyloidosis. To understand if trafficking T-cells also undergo clonal expansion, we analyzed the T-cell receptor (TCR) repertoire using scVDJ sequencing of CD8 T-cells from late and early disease as well as cognate Wt littermates, as in Fig. 1a. To assess clonality, TCRs were grouped based on clone size. In late disease, we observed markedly increased TCR clonality, with a higher proportion of expanded clones contributing to the overall repertoire. Notably, the Cd8_1 cluster contained many highly expanded clones (>200 cells) with the highest frequency of these expanded clones observed in the APP23-tg late group, as illustrated in Fig. 5e. T-cell expansion in Wt late was likely a consequence of aging-related inflammation. Differential gene expression analysis of clonotypes with >200 clones was highlighted by T-cell exhaustion markers when compared with the rest (Fig. 5f). A strong cumulative exhaustion score was, in turn, specific to late disease, underscored by upregulated Pdcd1, Lag3, and Tox, indicative of T-cell hyperactivation (Fig. 5g, h). Spatially, Pdcd1 that encodes exhaustion marker PD-1, was highly enriched in the plaque neighborhood (Fig. 5i, j), with exhausted T-cells strongly localized to the plaque edge (Fig. 5k), further attesting that hyperactivation leading to exhaustion as effector CD8 T-cells approach the plaque periphery is a phenotype of T-cell neuroinflammation in amyloid disease.

## ISG expression is plaque-associated in human amyloid pathology

Having defined a distinct Aβ-driven ISG response in our AD mouse model, we wanted to elucidate whether this response can also be observed contiguous to plaques in human AD. To this end, we subjected brain tissue cores from post-mortem patients with AD to targeted single-cell spatial transcriptomics using the Molecular

Cartography platform (Fig. 6a and Supplementary Tables 2 and 3). Aβ plaques were identified on assay sections, segmented, and plaque neighborhoods defined as "inside", "periphery" and "outside" (Fig. 6b). Expression of ISGs was then quantified as transcript densities in these ROIs per plaque. ISG expression was enriched "inside" the plaques for IRF7, IFIT1, and IFIT2, but USP18 showed a different trend with a subset of plaques having enhanced expression in the "periphery" but for others, the expression was not localized within the plaque neighborhood. Overall, the combined average expression of ISGs was enriched in the immediate plaque neighborhood, reflecting the observations from late disease in APP23-tg mice (Fig. 6c). In addition, at the transcript level, expression of CXCL10 transcripts was enriched in close proximity to plaques (Fig. 6d), further corroborating plaque-specific induction of CXCL10. Moreover, increased abundance of activation marker transcripts such as NKG7 and GZMK was indicative of T-cell activation, additionally reflected by high PDCD1 transcript density in the plaque neighborhoods (Fig. 6e, f). As PDCD1 is mostly expressed by exhausted T cells, we hypothesize that, similar to our pre-clinical observation, T-cells verging on plaques in human CNS amyloid pathology putatively undergo activation-induced exhaustion.

## Discussion

The fact that T-cells infiltrate the human AD brain was first described in 1988[20,21]. Over the last decade, the knowledge of T-cell abnormalities associated with AD and their contribution to disease progression has been chronicled in multiple mouse studies[26]. The patrol of CSF in AD patients by clonally expanded TEMRA cells further expanded the neuroinflammatory significance of the adaptive immune system[25]. However, the phenotype of focal adaptive immune response to Aβ hasn't been characterized in detail. To address this gap, we employed the APP23 mouse model, which enables the study of Aβ-driven T-cell responses without additional confounding neuropathological features. In summary, using single-cell transcriptomics, microscopy, and targeted spatial transcriptomics, we observed a localized T-cell response to Aβ, driven primarily by CD8+ T-cells that increased with progressing disease pathology. We discovered a subset of CD8+ T-cells with enriched ISG expression that was prominent in late disease that contributes to effector T-cell chemotaxis via the CXCL10-CXCR3 axis. Importantly, Type-I interferon signaling in CD8+ T cells in the context of amyloid pathology was first described by Altendorfer et al.[45]. Their bulk transcriptomic profiling revealed interferon-associated signatures in brain CD8+ T cells from APP–PS1 and aged WT mice, suggesting parallels to Trm-like responses observed in diverse CNS inflammatory conditions. Our work expands on these foundational observations by providing spatio-temporal resolution and identifying ISG CD8+ T cells as a late-stage, plaque-associated phenomenon. We show that the ISG response transitions from microglia in early disease to CD8+ T cells in late disease, revealing a dynamic shift in the cellular arbiters of interferon-driven inflammation.

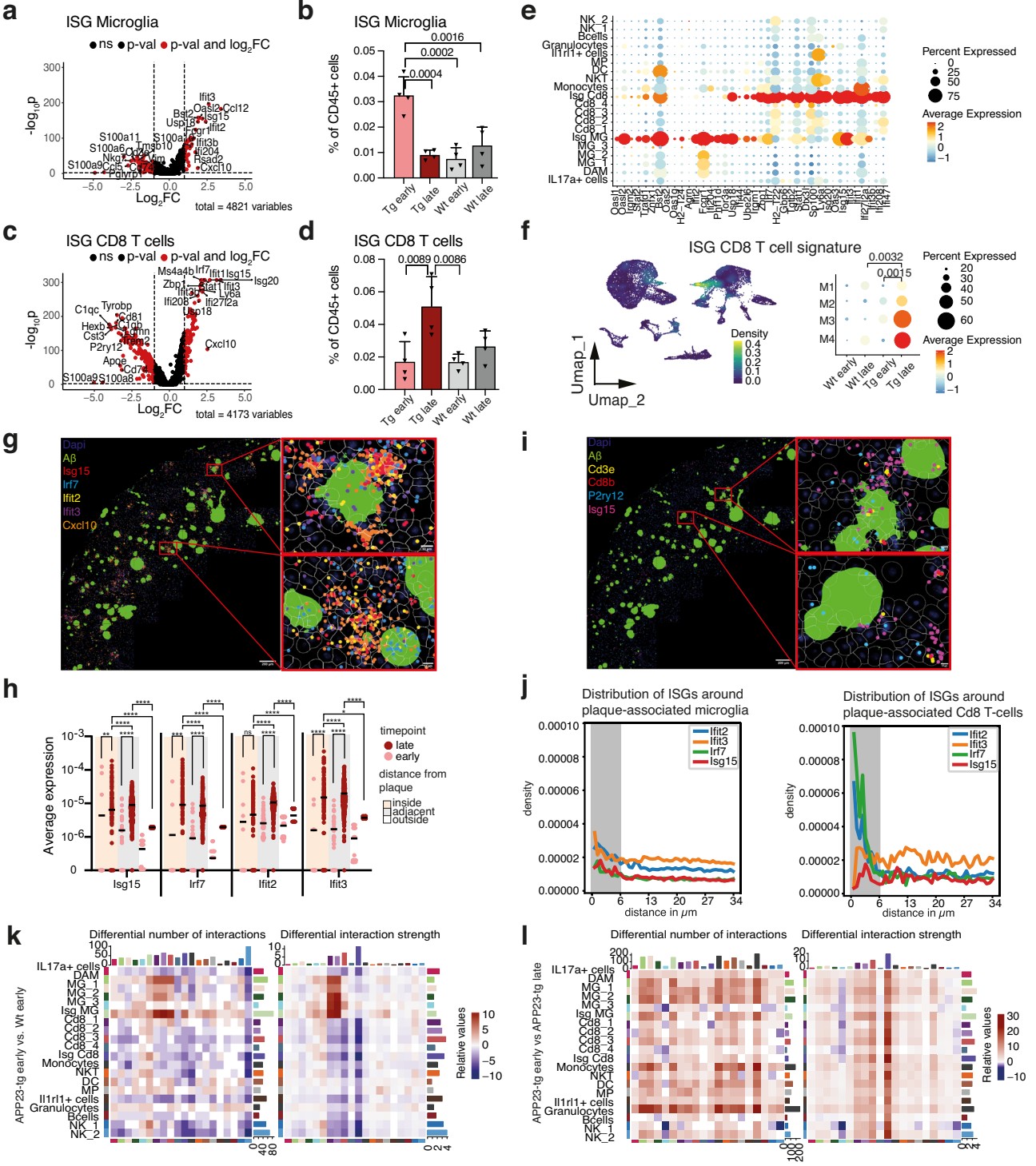

Fig. 3 | A distinct ISG T-cell subset drives Type-I interferon-mediated neuroin-flammation in late disease. a Volcano plot of differential gene expression (DGE) between ISG-expressing microglia and other immune subsets using the Wilcoxon rank-sum test. *P*-values: Benjamini-Hochberg (BH) adjusted. b Bar plot showing ISG+ microglia frequency within total CD45+ cells across cohorts (*n* = 4 mice/cohort). Statistics: two-way ANOVA. Data are presented as mean + SD. c Volcano plot of DGE between ISG+ CD8+ T cells and other immune subsets using the Wilcoxon rank-sum test (BH-adjusted *P*-values). d Bar plot showing ISG+ CD8+ T cell frequency within CD45+ cells (*n* = 4 mice/group). Statistics: two-way ANOVA. Data are presented as mean + SD. e Dot plot of average RNA expression and frequency for canonical ISGs across immune subsets. f Left: UMAP showing ISG CD8+ T-cell-specific ISG density. Right: Average RNA expression and frequency of the ISG CD8+ T-cell signature per mouse (M1-M4). g Spatial transcriptomic visualization of

selected ISGs in the cortex of a 24-month-old APP23-tg mouse. h Spatial quantifi-cation of ISG expression relative to plaque proximity in 12–14- and 22–24-month-old APP23-tg mice (*n* = 5). Regions: inside, adjacent (≤69 μm), and distant (>69 μm). Statistics: Kruskal−Wallis test with Dunn's correction (*p < 0.0332, **p < 0.0021, ***p < 0.0002, ****p < 0.0001). (i) Spatial maps of ISG+ CD8+ T cells and ISG+ microglia in a 24-month-old APP23-tg mouse (left: slide overview; right: zoom-in with segmented cells in white). j Spatial distance quantification between ISG expression and plaque-associated microglia (left) or T cells (right). k, l Heatmaps of inferred cell−cell communication (ligand−receptor interactions) showing interac-tion number (left) and strength (right) with signaling sources on the *y*-axis. Com-parisons include (k) APP23-tg vs. Wt (early stage) and l early vs. late stage (APP23-tg).

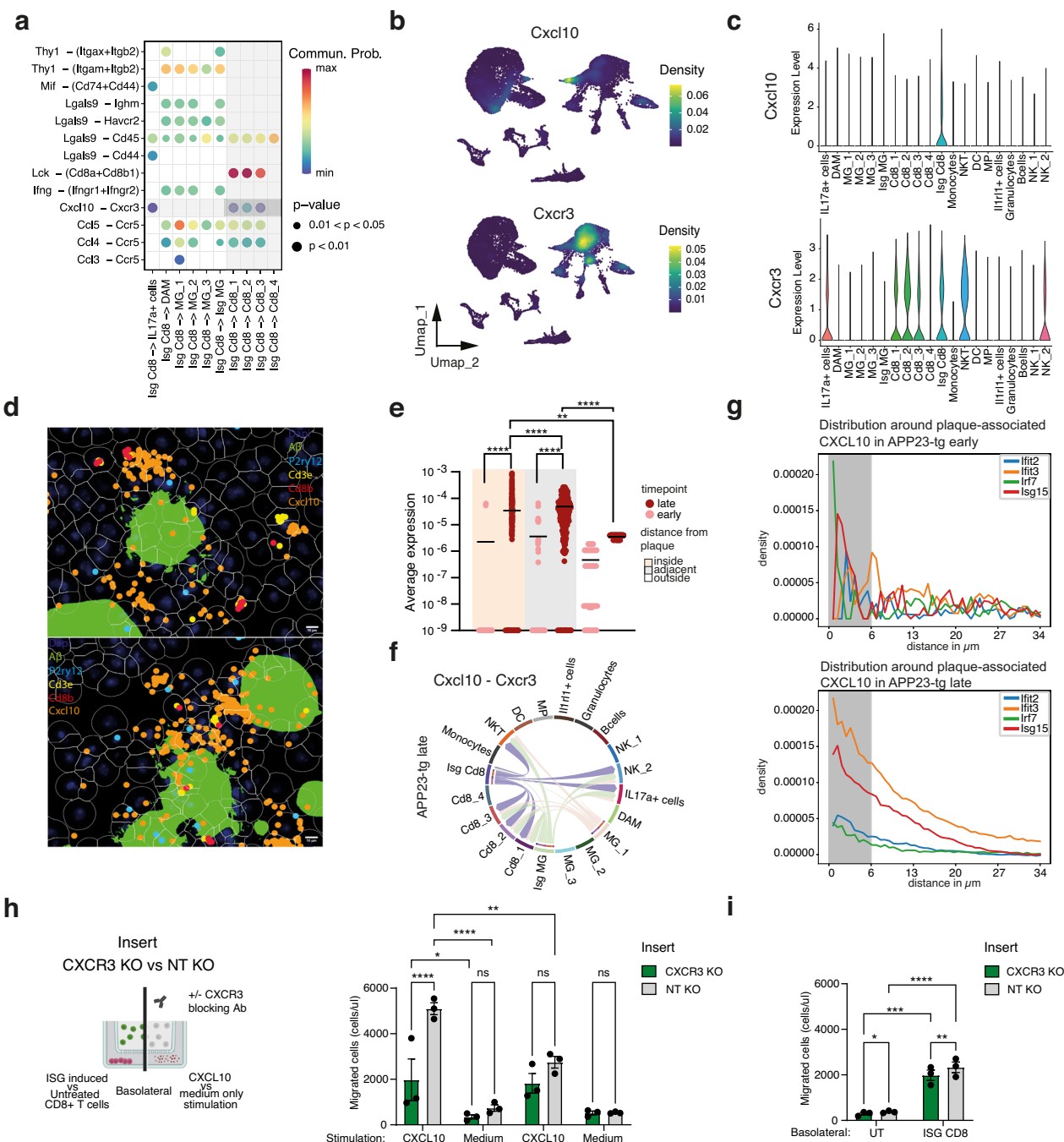

**Fig. 4 | ISG subset derived CXCL10 drives T-cell infiltration in response to Aβ. a** Differential analysis of enriched ligand–receptor interactions using ISG⁺ CD8⁺ T cells as the signaling source. Dot color: communication probability; dot size: p-value (one-sided permutation test, $p < 0.05$). **b** UMAP showing Cxcl10 and Cxcr3 RNA expression density across CD45⁺ immune cells. **c** Violin plots of average Cxcl10 and Cxcr3 expression across immune subsets. **d** Spatial RNA expression of Cxcl10 in representative regions of a 24-month-old APP23-tg mouse. **e** Spatial quantification of Cxcl10 relative to amyloid plaques: intraplaque, adjacent (≤69 μm), and distant (>69 μm) in 12–14 and 22–24 month-old mice ($n = 5$). Statistics: Kruskal–Wallis with Dunn's correction (*$p < 0.0332$, **$p < 0.0021$, ***$p < 0.0002$, ****$p < 0.0001$). **f** Chord diagram of Cxcl10-Cxcr3 signaling in late-stage APP23-tg mice. Bar size indicates total signal strength. **g** Spatial distribution of ISG marker expression near Cxcl10⁺ plaque-associated cells in early-stage (top) and late-stage (bottom) mice. **h** Transwell migration assays of CXCR3 knockout (KO) vs. non-targeting (NT) control CD8⁺ T cells. Left: schematic (Created in BioRender. D170, P. (2026) https://BioRender.com/ccvdxlm); right: bar graph of migrated cells/μl ± stimulation (CXCL10, medium, ± blocking antibody). Data: mean ± SEM; dots: biological replicates ($n = 3$ per cohort). **i** Quantification of migrated CXCR3 KO and NT CD8⁺ T cells toward untreated (UT) or ISG-induced CD8⁺ T cell supernatants. Data: mean ± SEM; dots: biological replicates ($n = 3$ per cohort). Statistics for (**h, i**): two-way ANOVA with Tukey's correction (*$p < 0.0322$; **$p < 0.0021$; ***$p < 0.0002$; ****$p < 0.0001$; ns = not significant).

Our observations align with recent findings of increased CD8+ T-cell accumulation in AD-like transgenic models and patients with AD. Moreover, correlates of plaque-associated ISG, CXCL10, and T-cell activation markers in human tissue samples support the relevance of local inflammatory Type-I interferon signatures to Aβ in driving compounding T-cell powered neuroinflammation. Being a model for amyloidosis, APP23-tg mice do have limitations in accurately modelling AD with the absence of features such as tauopathy. In this context,

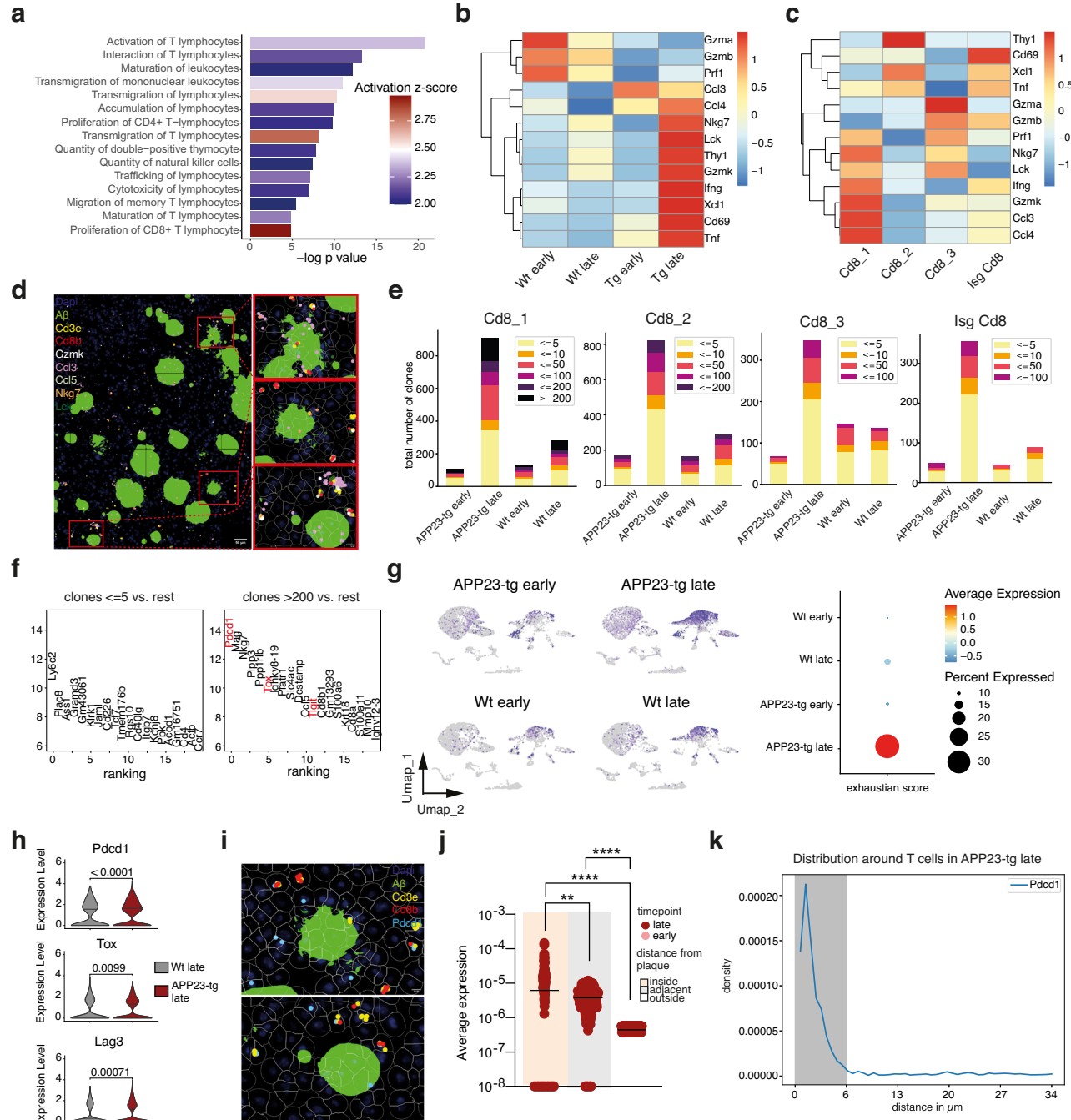

**Fig. 5 | Infiltrating CD8+ T-cells clonally expand and exhaust. a** Ingenuity Pathway Analysis (IPA) identifying the top upregulated immune signaling pathways in CD45+ immune cells from APP23-tg mice versus Wt controls. Significance was determined using a right-tailed Fisher's exact test and is shown as −log₁₀(*p*-value). Pathways are ranked by significance (*p*-value) and color-coded by activation z-score. **b** Heatmap showing normalized average RNA expression of selected T-cell activation markers across all cohorts. Color intensity indicates relative expression levels. **c** Heatmap showing normalized RNA expression of the same activation markers specifically within CD8+ T-cell subsets. **d** Spatial transcriptomic maps of T-cell activation markers in 24-month-old APP23-tg mice. Whole-slide views are shown on the left, with zoomed-in regions of interest on the right. **e** Number of T-cell clones grouped by clone size (visualized by color gradient) in identified T cell subsets between all cohorts. **f** Z-score depicting upregulated genes from differential expression analysis in non-expanding clones (absolute clone numbers ≥ 5) and highly expanding clones (absolute clone numbers >200). **g** Left: UMAP plot of

CD45+ immune cells overlaid with an expression density gradient of a defined T-cell exhaustion gene signature (Pdcd1, Lag3, Tox). Right: Average RNA expression and expression frequency of the exhaustion signature across all cohorts. **h** Violin plots comparing average expression levels of exhaustion-associated genes between late-stage APP23-tg and wild-type mice. Statistical significance was assessed using the Wilcoxon rank-sum test. **i** Spatial transcriptomic visualization of Pdcd1 (PD-1) expression in two selected regions of interest from 24-month-old APP23-tg mice. **j** Quantification of Pdcd1 RNA expression in relation to plaque proximity: intra-plaque, plaque-adjacent (≤69 μm from plaque boundary), and plaque-distant (>69 μm). Analysis was performed in 22–24-month-old APP23-tg mice. Statistical testing was performed using the Kruskal–Wallis test with Dunn's correction for multiple comparisons; significance levels are shown as *= *p* < 0.0332, **= *p* < 0.0021,***= *p* < 0.0002,****= *p* < 0.0001) (**k**) Spatial distribution of Pdcd1 expression relative to T-cell markers in a 24-month-old APP23-tg mouse, visualized by density plots highlighting proximity to T-cell-enriched regions.

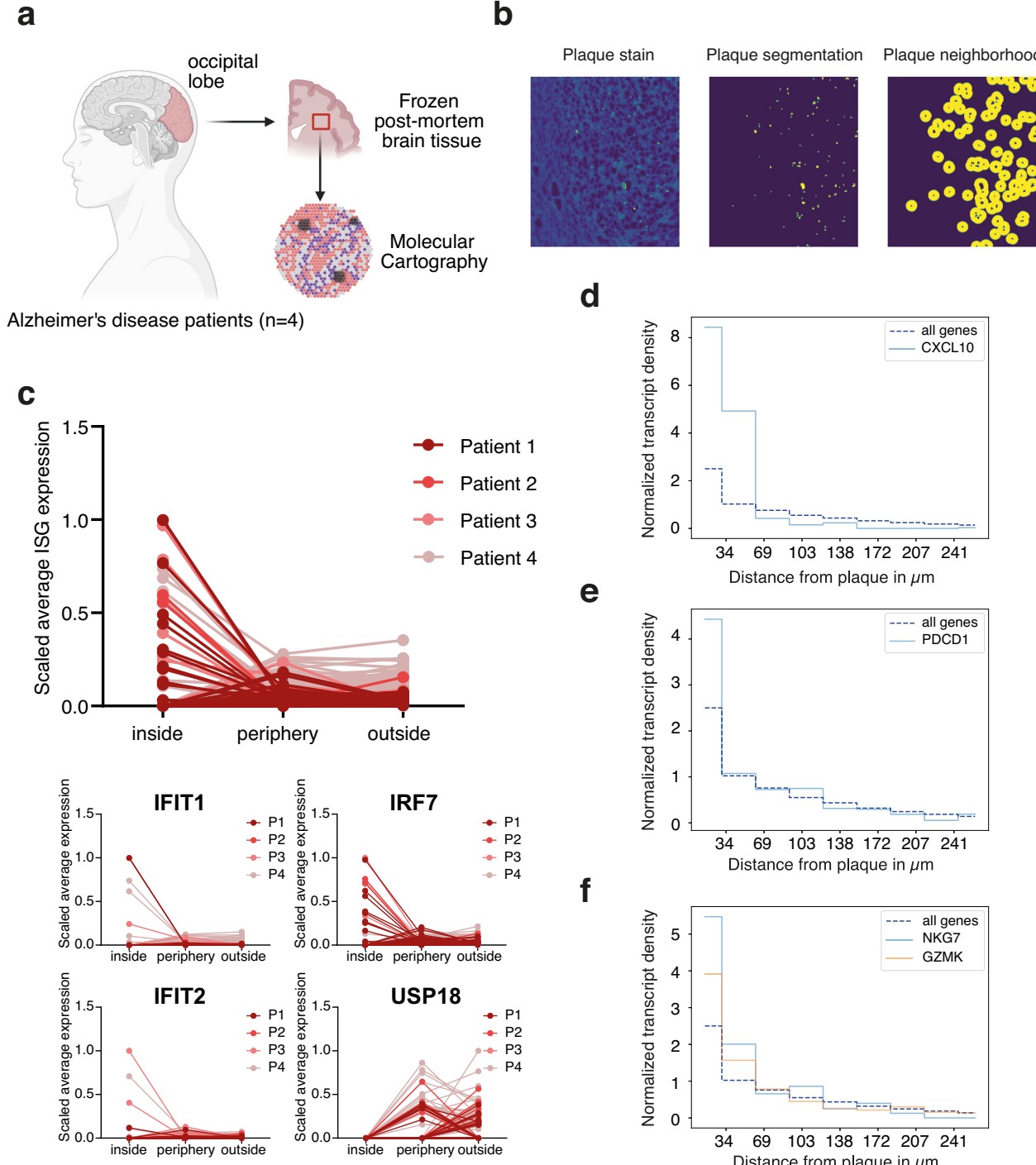

**Fig. 6 | ISG and CXCL10 expression localizes to Aβ plaque neighborhoods in human amyloid pathology. a** Schematic sketch of human material selected for targeted spatial single-cell transcriptomic assay conducted with the Molecular Cartography platform of Resolve Biosciences (Created in BioRender. D170, P. (2026) https://BioRender.com/nxb2lb3). **b** The workflow of plaque segmentation and identification of plaque-specific neighborhoods is depicted in one exemplary section. **c** Combined average (upper panel) and individual (lower panel) transcript density per plaque of indicated ISG markers identified in the spatial single-cell transcriptomic assay. Data was scaled and plotted for each patient separately. P = Patient. Normalized transcript density as a function of distance from plaques for **d** *CXCL10*, **e** *PDCD1*, **f** NKG7, and GZMK compared to average transcript density of all other genes.

Chen et al. recently showed an increase in the number of cytotoxic T-cells in areas with tau pathology in mice with tauopathy expressing human apolipoprotein E (APOE) isoforms and the dynamic transformation of T-cells from activated to exhausted states[28]. Interestingly, however, this phenomenon was absent in tau mice lacking APOE, indicating the stronger immunomodulatory relevance of APOE as compared to tau. APOE, especially APOE4, is a strong genetic risk factor for AD, is more closely associated with amyloid pathology, and critically regulates Aβ accumulation[46,47].

Studies investigating the impact of T-cell depletion (total or subset-specific) in controlling neurodegenerative processes have yielded conflicting results, ranging from increased Aβ plaque burden and cognitive impairment[48,49] to increased amyloid clearance, reversal of cognitive decline, and improved neuronal function[26,50] depending on

the study. These variances were most likely due to different models used, sub-populations targeted, as well as the time point of disease development studied. In most cases, the use of genetically modified models lacking T-cells or early depletion elicited detrimental pathological effects, whereas, ablation after fully developed amyloid pathology steered towards protective effects. Since our findings show that T-cell states and their proportion change with disease stage, longitudinal depletion studies in the future will help better understand when the T-cell influx is neuroprotective or pathological and what triggers drive the switch.

Although we observed an overall increase in the localization of T-cells in the plaque neighborhoods with disease course, T-cells had a greater propensity for parenchymal plaques. This reflected observations in a previous study where the number of CD8+ T-cells in the parenchyma of post-mortem AD samples positively correlated with Braak stages but not with age[26], with no such correlation observed for vascular CD8+ T-cells and Braak stages or age. This propensity may be a result of differing milieus between perivascular niches and the parenchyma. Characterizing these differences was not within the scope of this study, but warrants further investigations, as it may shed light on neuroinflammatory processes specific to CAA. As early interferon response manifests through ISG microglia, it would be of interest to ascertain whether induction of ISG CD8+ T-cells happens via interaction with ISG microglia or through initial direct interaction of CD8+ T-cells with Aβ. Microglia-specific IFN-receptor 1 (IFNAR1) deletion reduced ISG expression and Stat1 levels in plaque-associated microglia and rescued post-synaptic loss in the 5xFAD model[15]. Such a system would be well suited to mechanistically characterize how ISG CD8+ T-cells originate.

The role of CXCR3 in modulating T-cell trafficking has been well-established[51], with the CXCL10-CXCR3 axis additionally being important in influencing CD8+ T-cell response during chronic infection[52] and in AD[53,54]. Blockade of CXCR3 using a neutralizing antibody in a human neuroimmune axis model inhibited T-cell infiltration and attenuated neurodegeneration[53], and CXCR3 deficient APP/S1 mice exhibited reduced plaque burdens and diminished behavioural deficits[55]. The former study had observed elevated levels of astrocytic CXCL10 in the 5xFAD model, which was not replicated in our analyses. In this study, however, we identify ISG CD8+ T-cells as the source of CXCL10 in the Aβ plaque local inflammatory milieu, which drives infiltration of CXCR3+ CD8+ T-cells with a cytotoxic potential that clonally expand upon activation in the plaque neighborhood and fates to exhaustion. Our functional assays substantiate this proposed CXCL10-CXCR3 chemotactic mechanism. Using CRISPR-mediated CXCR3 knockout in primary T cells, we show that CXCR3 is required for efficient migration toward CXCL10 gradients. Moreover, ISG-induced CD8+ T cells act as a potent source of chemotactic cues: they drive strong transmigration of control T cells, whereas migration of CXCR3-deficient cells is markedly reduced. Together, these experiments bridge transcriptional, spatial, and functional evidence and support a paracrine model in which ISG+ CD8+ T cells recruit CXCR3+ effector T cells to amyloid lesions. Although additional chemokine-receptor pairs may contribute, our data identify CXCL10-CXCR3 signaling as a mechanistically relevant driver linking local type-I interferon activation to adaptive immune cell infiltration in amyloid pathology.

Although the blockade of CXCR3 sounds promising as a therapeutic strategy, strategies targeting ISG CD8+ T-cells may offer more specific alternatives for controlling neuroinflammation. Blockade of IFN-I receptor (IFNAR) in pre-clinical studies rescued memory and synaptic deficits and reduced microgliosis and inflammation[11,15]. Such an IFNAR1-blocking antibody – Anifrolumab, has been clinically tested in patients with systemic lupus erythematosus, where treatment downregulated multiple type-I IFN-induced pathways, downstream chemokines associated with immune cell migration, and proinflammatory cytokines[56]. Another drug, Ruxolitinib, which is a small-

molecule Janus kinase (JAK) inhibitor, FDA-approved for the treatment of inflammatory disorders, has very recently been reported to attenuate microgliosis and expression of proinflammatory cytokines, highlighting its potential in treating neuroinflammation[57]. Understandably, such strategies emphasize the necessity to perform longitudinal assessments of T-cell phenotypes in human amyloid pathology samples to decipher the peak abundance of ISG CD8+ T-cells in CNS vasculature and brain parenchyma.

As our time-resolved study suggests that Type-I interferon response translates over time from ISG microglia to ISG CD8+ T-cells, it will be of high relevance to study T-cell phenotypes during antibody-based removal of Aβ plaques. Anti-Aβ treatment was shown to reduce markers of amyloid in early AD and moderately reduce cognitive decline compared to placebo, but was associated with amyloid-related imaging abnormalities (ARIA) with edema or effusions (ARIA-E) and sulcal/leptomeningeal hemosiderin deposits (ARIA-H)[58]. As there is a perspective in the field that rapid Aβ disaggregation at high doses of anti-Aβ immunotherapy exaggerates the inflammatory response of vessel-adjacent immune cells that can disturb the integrity of the cerebrovascular wall, resulting in ARIA[59], there is a high probability that local adaptive immune responses are altered upon treatment. It is tempting to speculate that a shift from ISG microglia to T-cells would support the disintegrity of the perivascular niche, leading to ARIA. This may also be the case for spontaneous ARIA-E in CAA-related inflammation (CAA-ri), which is associated with increased levels of anti-Aβ autoantibodies in the CSF and focal increase of microglial activation. As the magnitude of microglial activation associated with ARIA-E and ARIA-E severity was observed to be influenced by the coexistence of CAA and AD[60], it is possible that, in the context of anti-aβ immunotherapy, exacerbation of background inflammatory response to CAA, facilitating a more pronounced T-cell response, could drive higher frequency and severity of ARIA. Of note, in the Lecanemeb study, the frequency of ARIA was higher among APOE4 carriers; however, in a study of CAA-ri, APOE4 was not a prognostic predictor of outcome but prognosis rather alluded to the nature of the focal inflammatory response[61]. Dysregulation in the peripheral blood CD4+ T-cell sub-population mediated by APOE4 genotype has been recently shown to be associated with neuroinflammation in AD[62]. Evaluation of how APOE4 influences peripheral CD8+ T-cells will elevate our understanding in the future of the time-resolved induction of ISG CD8+ T-cells in the plaque-burdened brain and may provide peripheral biomarkers for Type-I interferon neuroinflammatory response.

Overall, our study expands on the knowledge of inflammatory cascades in response to Aβ, highlights the relevance of Type-I interferon response and associated CD8+ T-cells in neuroinflammation, and suggests its time-restricted targetability. Future studies targeting Type-I Interferon response will therefore be of pertinence in understanding how broadly it controls neuroinflammation and reverses neurodegenerative pathologies.

## Methods

### Informed consent and ethics committee approvals
All animal procedures followed the institutional laboratory animal research guidelines and were approved by the governmental authorities (Regional Administrative Authority Karlsruhe, Germany). Human tissues were used after approval of the local regulatory authorities (Ethics Committee at the Medical Faculties of the University of Heidelberg).

### Mice
APP23-tg mice contain a human amyloid precursor protein (APP751) cDNA with the Swedish double mutation at position 670/671 under the control of the neuron-specific Thy-1 promoter (Calhoun et al., 1999). Heterozygote B6, D2-TgN[Thy-APPSWE]−23- tg mice (APP23) provided by Matthias Staufenbiel (Novartis Institutes for BioMedical Research,

Novartis Pharma AG, Basel, Switzerland) were backcrossed twice with C57BL/6 mice (Janvier, Saint Berthevin Cedex, France) and kept under a 12/12 h light/dark cycle with standard food and water ad libitum. Wild-type (WT) littermates were used as control animals.

The Gt(ROSA)26Sor^tm1.1(CAG-cas9*,-EGFP) Fezh mouse line was acquired from Jackson Laboratory. It carries a Cas9-EGFP construct inserted into the Rosa26 locus, allowing ubiquitous expression of Cas9 and EGFP under the control of the CAG promoter. Mice were housed under specific pathogen-free conditions with a 12 h light/12 h dark cycle, at an ambient temperature of $22 \pm 2\,°C$ and relative humidity of 50–60%, with ad libitum access to food and water.

### Human Alzheimer's disease tissues

Cryo-conserved post-mortem Alzheimer's disease patient tissue was obtained from the Neurobiobank Munich (Center for Neuropathology and Prion Research, Munich). All participants gave prospective pre-mortem written consent for their brains to be banked and used for research. Patient demographics and information on disease staging are summarized in Supplementary Table 3.

### Immunofluorescence of formalin-fixed paraffin-embedded tissue

APP23-tg mice were perfused with 4% formalin (RotiHistofix 4%, Roth) under isoflurane anaesthesia. Brains were dissected and fixed for a maximum of 24 h in 4% formalin at $4\,°C$. Coronal tissue slides of the cortex were cut with a thickness of $7\,\mu m$ using a microtome (Thermo Scientific). For immunofluorescence stainings, paraffin was removed by incubating the slides at $37\,°C$ for 5 min and placing them into two cups of xylol for 10 min each. To rehydrate sections, they were transferred to two containers of 100% Ethanol for 5 min each, and a decreasing Ethanol-row (96%, 70%, and 50%) was performed for 3 min each. Slides were collected in VE water. For antigen retrieval, slides were transferred into a plastic cuvette, sealed with parafilm, and boiled in Cell Conditioning Solution pH 6.0 (Ventana, Roche Diagnostics) for 30 min in a food steamer. Sections were cooled down for 20 min in the same plastic cuvette and washed 3 times in washing buffer (0.1% Tween20 in 1x PBS). Immunofluorescence stainings were conducted sequentially, meaning that primary antibodies for different targets were not mixed but incubated one after another together with their matching secondary antibody.

The tissue was blocked with the appropriate serum, matching the species of the secondary antibody (4% goat or donkey serum in washing buffer) for 1 h at RT. Slides were incubated overnight at $4\,°C$ with the respective primary antibody targeting CD3 (polyclonal rabbit anti-mouse/human, DAKO A0452, 1:200), CD31 (polyclonal rabbit anti-mouse, Abcam 28364, 1:100), or amyloid beta (monoclonal mouse anti-mouse, BioLegend 803014, 1:200) in blocking buffer. The next day, slides were washed 3 times with washing buffer before they were incubated with fluorescently labelled secondary antibodies to target CD3 (polyclonal goat anti-rabbit AF633, Thermofisher A-21070, 1:200), CD31 (polyclonal goat anti-rabbit AF546, Thermofisher A11010, 1:500), or amyloid beta (polyclonal goat anti-mouse AF488, Thermofisher A11029, 1:1000) for 1 h at RT. After thorough washing, sections were mounted using VECTASHIELD HardSet Mounting Medium with DAPI, which also stains Amyloid beta plaques. Images were acquired on an LSM700 confocal microscope (Zeiss) with 20x magnification.

### Immunofluorescence of fixed frozen tissue

APP23-tg mice and Wt littermates were perfused with ice-cold PBS under ketamine/xylazine anaesthesia (120 mg/kg ketamine + 20 mg/kg xylazine in sterile 0.9% NaCl solution, 0.1 ml/10 g body weight). Brains were dissected and snap-frozen in Tissue-Tek® (Sakura). Coronal tissue slides of the cortex were cut with a thickness of $7\,\mu m$ using a microtome (Thermo Scientific). For immunofluorescence stainings, the tissue was blocked with the appropriate serum, matching the species of

the secondary antibody (4% goat or donkey serum in washing buffer) for 1 h at RT. Slides were incubated overnight at $4\,°C$ with the respective primary antibody targeting CD3 (polyclonal rabbit anti-mouse/human, DAKO A0452, 1:200), CD31 (polyclonal rabbit anti-mouse, Abcam 28364, 1:100), or amyloid beta (monoclonal mouse anti-mouse, BioLegend 803014, 1:200) in blocking buffer. The next day, slides were washed 3 times with washing buffer before they were incubated with fluorescently labelled secondary antibodies to target CD3 (polyclonal goat anti-rabbit AF633, 1:200), CD31 (polyclonal goat anti-rabbit AF546, 1:500), or amyloid beta (polyclonal goat anti-mouse AF488, 1:1000) for 1 h at RT. After thorough washing, sections were mounted on day 4 using a mounting medium with DAPI. Images were acquired on an LSM700 confocal microscope (Zeiss) with 20x magnification.

### T-cell quantification

T-cells were quantified at 8, 12, 16, 20, and 24 months of age in cortices of APP23tg mice. Formalin-fixed paraffin-embedded tissue was used, and T-cells were stained as previously described. Three mice per age and three brain slides per mouse were used. Three tile scan images of an area of $960.25\,\mu m^2$ were acquired for each slide, and T-cells were counted using an ImageJ macro. The background was subtracted from both the DAPI and Alexa Fluor 633 (AF633) channels, using the "Rolling Ball Background Subtraction" with a radius of 30 pixels. Subsequently, the images were filtered using the "Gaussian Blur" function with the radius set to 5. DAPI images were segmented using the "Find Maxima" tool with the prominence set to 1, and the threshold set accordingly. The output of Find Maxima was single points highlighting the centre of each DAPI-stained nucleus. The threshold for a true positive signal in the AF633 channel was set individually, and an area of the red signal was created. All single points marking DAPI maxima that were detected within the selected red area were counted as T-cells, whereas, all DAPI maxima were counted as total cells. For statistical analysis of T cell infiltration, mean values of each slide per time point were used ($n = 9$). Data were checked for normality using Prism's built-in D'Agostino & Pearson test, and changes in T cell numbers normalized to total cells were calculated by One-way ANOVA and Tukey's post hoc test. The significance level was set to 0.05.

Parenchymal and vascular amyloid beta plaques were counted manually on each tile scan, utilizing the "cell counter" function in Fiji. For statistical analysis of total plaque burden, mean values of each slide per time point were used ($n = 9$). Data were checked for normality using Prism's built-in D'Agostino & Pearson test, and changes in plaque numbers were calculated by One-way ANOVA and Tukey's post hoc test. The significance level was set to 0.05.

Spearman correlation analyses were run in Prism and plotted in linear regression models. Therefore, all values, independent of the age group, were used as single data points for correlation analysis. 8-month-old mice do not show any plaque burden, and 12-month-old mice only in very few slides. Since there is a baseline abundance of T-cells in the brain, these two-time points were excluded from the "T cell/plaque" analyses, as this would have led to a skewed picture. T-cells surrounding either parenchymal or vascular amyloid deposits were counted manually on a set of well-defined plaques of both types to prevent double counting of T-cells. Therefore, definite plaques were marked, and a circle with a radius of 90 pixels was drawn around the center. T cells falling into the defined area were counted, and ratios of T-cells per parenchymal and vascular amyloid deposits were calculated.

### Isolation of brain-associated immune cells

APP23tg/APP23wt mice were sacrificed by intraperitoneal overdose of Ketamin/Rompun; brains were isolated without prior perfusion to capture a comprehensive representation of the immune micro-environment, including both resident brain and infiltrating immune cells, followed by digestion with 50 ug/ml Liberase (Sigma) at $37\,°C$ for

8 min and mechanical dissociation through 100 and 70 μm strainers (Miltenyi) to generate a single cell suspension. Subsequently, Myelin removal was performed using a 30% continuous Percoll (GE Healthcare) gradient and centrifugation at 2700 rpm.

### Fluorescence-activated cell sorting of mouse brain immune cells and library preparation for single-cell RNA- and VDJ-seq

After preparation of the single cell suspension, staining for Fluorescence-activated cell sorting (FACS) was performed. Murine cells were blocked with anti-mouse CD16/32 (0.5 μg per well, eBioscience) and stained for CD45 (BV510, Biolegend 103138), CD3 (APC, Biolegend 100236), and CD11b (FITC, Biolegend 101206). In addition to that, eFluor 780 fixable viability dye (eBioscience) was used according to the manufacturer's protocol to exclude dead cells. Furthermore, for single-cell RNA sequencing, titrated amounts of Total-Seq C hashtag antibodies (TotalSeq™-C0301 anti-mouse Hashtag 1 Antibody, Biolegend 155861; TotalSeq™-C0302 anti-mouse Hashtag 2 Antibody, Biolegend 155863) were added to the cell-antibody mix in a total of 200 μl and incubated for 30 min on ice. Leukocytes were sorted for sorted for subsequent steps using a CD45+ s. Cells were sorted on BDAria II through a 100 μM nozzle and 4-way purity. Cells were sorted in 5 μl 0.04% BSA in PBS and kept on ice until processing. Single-cell capture, reverse transcription, and library preparation were carried out on the Chromium platform (10x Genomics) with the Single Cell 5′ reagent v2 kit (10x Genomics) according to the manufacturer's protocol using 8000–23,000 cells as input per channel. Each pool of cells was tested for library quality, and library concentration was assessed. Each of the final libraries was paired-end sequenced (26 and 92 bp) on one Illumina NovaSeq 6000 S2 lane.

### Single-cell transcriptomic analyses

**Quality control and normalization.** Single-cell RNA data were processed using the CellRanger pipeline (version 5.0) to the GRCm38 reference genome with all default settings. All downstream analyses were performed using Seurat v 4.4.0. All cells that had unique feature counts over 4000 (late time point)/3000 (early time point) or less than 200 (early and late time point), as well as cells with more than 5% mitochondrial counts, were excluded from downstream analyses. The cells of each mouse were stained with two different TotalSeq C hashtag antibodies during FACS staining. Using the HTODemux() command with default settings, following Seurat's demultiplexing vignette (https://github.com/satijalab/seurat/blob/master/vignettes/hashing_vignette.Rmd), cells were demultiplexed to their respective sample of origin. Cells without an assigned hashtag (unassigned cells) or doublets (cells with two or more assigned hashtags) were removed from the analysis. Subsequently, only cells classified as singlets (67.8%) were used for further downstream analyses. Visualization of demultiplexing results, as well as unassigned cell/doublet proportions for each mouse, can be found in the Supplementary material section. Furthermore, mice of each group, as well as early and late time points, were merged, and a master object was created. Gene expression data were scaled and normalized using regularized negative binomial regression as implemented in ScTransform(). In addition to that, 2000 highly variable genes were selected also using ScTransform(). VDJ data from the filtered contig annotations files were added to the RNA data after running the cellranger 5.0 VDJ pipeline.

**Integration of different groups and clustering.** Subsequently, cells from all 16 mice were integrated using Seurat's FindIntegrationAnchors() and IntegrateData() with standard parameters and ndims = 20. The data was scaled, and a principal component analysis was done using RunPCA() with npcs = 30. 20 principal components were selected based on the inflection point in the elbow plot. The generated dimensions were used to cluster cells by running FindNeighbours(), FindClusters(), and RunUMAP(). Umap was visualised using a

resolution of 1.0. Differential gene expression analysis was performed using MAST (v1.26.0) to determine the identity of each cluster, and highly upregulated genes were used to label each cluster manually. To validate the cluster annotation, normalized expression values of 830 microarray samples of pure mouse immune cells from the Immunologic Genome Project (ImmGenData) were obtained using the celldex package (v1.10.0) and mapped onto the Umap using the SingleR package (v2.4.1).

**Receptor-ligand interaction analysis.** Receptor ligand analysis was performed using the cellchat package (v1.6.1). First, the dataset was split into four different groups (APP23-tg early/late, APP23-wt early/late), followed by creating cellchat objects for each of these groups using createCellChat() with default parameters. Transformed and normalized gene expression data were used as input, as well as the cluster annotation for defining the interacting cell groups. Subsequently, the CellChat database CellChatDB for murine analyses was obtained, containing 2021 validated molecular interactions. Overexpressed ligands and receptors were identified by running identifyOverExpressedGenes() and identifyOverExpressedInteractions(), and communication probability/strength between any interacting cell groups was calculated using computeCommunProb() as well as computeCommunProbPathway(). Lastly, to identify dominant senders, receivers, mediators, and influencers in all inferred communication networks netAnalysis_computeCentrality() was computed.

**Ingenuity pathway analysis.** QIAGENs Ingenuity Pathway Analysis (IPA) bioinformatic tool (accessed 2024) was used for downstream analyses and visualization of the scRNAseq dataset. The parent Seurat object was divided into individual objects consisting of single-cell clusters. For all CD45+ cells, differentially expressed (DE) genes between conditions (APP23tg versus Wt) or time points (late versus early) were identified by the FindMarkers() function that is based on a two-sided Wilcoxon rank-sum test. Average log2(fold change) values of DE genes were then supplied to IPA. To obtain heatmaps of "Diseases and functions" and "Canonical Pathways", genes were filtered for mice as the species, immune cells as the tissue, and for a right-sided p-value < 0.05 by IPA's implementation of Fisher's exact test.

**Single-cell T-cell receptor analysis.** The T-cell receptor analysis was done using the scanpy (v1.10.1) and scirpy (v0.17.0) packages. The clustered data object from Seurat was loaded into a MuData object together with the receptor sequencing data. The data was subset to the CD8+ T-cells, and the clonal expansion was defined using the scirpy function tl.clonal_expansion() and analyzed for the subclusters. Differential gene expression analysis between the different clonal expansions was performed using the scanpy tl.rank_genes_groups() function.

### Targeted spatial transcriptomics

**Sample preparation.** Coronal sections of the cerebral cortex of in total eight cryo-preserved APP23-tg and Wt mouse brains (n = 2 per group) were subjected to targeted spatial transcriptomics using the Molecular Cartography™ platform (Resolve Biosciences). For each mouse, a single brain hemisphere was analyzed. A coronally adjacent section was stained with pFTAA dye to detect Aβ deposits and guide ROI selection within the cerebral cortex for transcript detection. The list of transcripts probed is provided in Supplementary Table 1.

For human post-mortem Alzheimer's disease patient tissue from the cerebral cortex, 1 cm³ cryo-conserved tissue cores were sectioned. Two sections per patient were used for spatial transcriptomics using the same platform as murine tissue. An immediately adjacent section was stained with Thioflavin S to detect Aβ deposits and guide ROI selection within the cerebral cortex for transcript detection. The list of transcripts probed is provided in Supplementary Table 2.

**Methodology of Molecular Cartography platform.** Molecular Cartography is an imaging-based, highly multiplexed single-molecule fluorescence in situ hybridization (smFISH) technology designed to detect and quantify individual RNA transcripts with subcellular resolution while preserving tissue morphology. The technique works in principle by employing a proprietary system of combinatorial barcoded probes and sequential imaging rounds. Briefly, custom designed, gene-specific oligonucleotide probes, each bearing a unique barcode, are hybridized to target RNA molecules within tissue sections. In subsequent iterative imaging cycles, these barcodes are decoded through the sequential binding and unbinding of fluorescent reporter molecules. Each cycle involves the detection of specific fluorophores associated with different barcode bits. By recording the fluorescent signal across multiple rounds, a unique "fingerprint" is generated for each RNA molecule, allowing its identification and precise spatial localization. This imaging-based method achieves subcellular resolution (approximately 300 nm), enabling the visualization of individual transcripts and their spatial context within cells and tissues. For our experiments, tissue sections were prepared according to Resolve Biosciences' guidelines and processed in their service laboratory. Following data acquisition, transcripts were segmented to individual cells based on DAPI nuclear staining and Resolve Biosciences' proprietary cell segmentation algorithm. Further information on the methodology is available on the company's website as well as in the following publication[63].

**Mouse spatial transcriptomic analysis.** For the cell segmentation-free analysis of the mouse spatial transcriptomics data, the raw data provided by the Molecular Cartography platform of Resolve Biosciences was loaded into a pandas dataframe. Additionally, the respective pFTAA stained images were imported using the tifffile package, normalized, and segmented via a threshold. Then the distance of each marker to the closest plaque was calculated using scipy.ndimage.distance_transform_edt(). The transcripts were then assigned to three categories, inside ($d \leq 0\,\mu$m), plaque adjacent ($0 < d <= 69\,\mu$m, corresponding to 500px on rendered images of assay slides), and outside ($d > 69\,\mu$m). The colocalization of two markers was measured by individually calculating the histogram of the distance of marker 1 to a single transcript of marker 2. Then the histograms for all occurrences of marker 2 were averaged and renormalized by the annulus size of the respective bin $\pi$ (bin2-(bin-1)2). To measure the colocalization of markers in tissue-adjacent regions, marker 2 occurrences were filtered to be only from the inside and plaque-adjacent regions.

The association of single markers with cell types and the colocalization of the marker in question were determined for a set of marker genes, where the colocalization was calculated for each marker gene (set as marker 2), and the mean was plotted. T-cell markers used were Cd3e, Cd4, Cd8b1, Foxp3, Trbc, Trdc, Cd163l1n and Ccr7; the microglia markers used were Cst3, Hexb, Tyrobp, Ctsd, P2ry12, Tmem119, Tgfbr1, Olfml3, Cst7, Apoe, Lpl, Trem2, Srgap2, and Nav2.

For the cell segmentation-based analysis of the mouse spatial transcriptomics data, segmented cells from the mouse samples were loaded in one scanpy AnnData object. The number of available markers varied across the samples. The markers were filtered to keep only markers with less than 60.000 missing values; the remaining missing values were set to 0. This resulted in a dataset with 97,596 cells and 109 markers. The cells were filtered to have a minimum of 20 counts expressed. Normalization was performed using Pearson residuals with sc.experimental.pp.normalize_pearson_residuals(), on the resulting residuals, the principal components were calculated and integrated with harmony (v0.0.10). The resulting integration was then clustered with Leiden clustering and visualized with a UMAP embedding (Fig. S6).

The gene panel was designed to differentiate cell types by including cell type markers, the marker sets for the individual cell types were scored for each cell with the sc.tl.score_genes() function (Fig. S6). Next, the Leiden clusters were assigned a cell type if the mean score for a cell type was significantly higher than in all other cells. Based on the following condition, for each Leiden cluster (lc) and cell type (ct), (mean(score$_{ct}$ (lc))-mean(score$_{ct}$(other cells)))/ (std(score$_{ct}$(lc))+std(score$_{ct}$(other cells)))>1, the clusters were annotated. This score uniquely assigned cell types to clusters, some clusters could not uniquely be annotated (Fig. S6).

Next, the data was subset to the T-cells, and PCA, harmony integration, and Leiden clusters were recomputed. To identify exhausted and ISG T-cells, the cells were scored for the respective gene sets: ISG genes: Ifit1, Ifit2, Irf7, Ifnar1, Usp18, Isg15, Cxcl10; exhaustion genes: Pdcd1, Havcr2, Lag3, Tigit, Ctla4, Cd244, Cd160, Cd96, Cd38, Tox. Based on these scores, two clusters were annotated as ISG T-cells, and one cluster was annotated as exhausted T-cells.

The distance of each cell to the plaques was calculated as previously described, and the histograms of the annotated cell types with respect to the distance from the plaque were plotted. The values of the histograms were renormalized to respect the quadratically increasing area covered by the bins.

**Human spatial transcriptomics analysis.** For the transcript-wise analysis of the spatial transcriptomics data, the raw output data describing transcript positions provided by the Molecular Cartography platform of Resolve Biosciences was loaded into a pandas dataframe. Additionally, the respective Thioflavin S-stained images were imported using the tifffile package, normalized, and segmented via a threshold. Then the distance d of the transcripts to the closest plaque was calculated using the scipy.ndimage.distance_transform_edt() function. The transcripts were then assigned to one of three categories, inside the plaque ($d \leq 0\,\mu$m), plaque adjacent ($0 < d \leq 69\,\mu$m, corresponding to 500px on rendered images of assay slides), and outside ($d > 69\,\mu$m). Then the transcript density for each marker and each of the three categories was calculated by normalizing the number of transcripts with the size of the region. This transcript density was then plotted for each marker for the three regions.

Furthermore, histograms of the transcript density surrounding the plaques were generated with numpy.histogram(density=True). The histogram values were then renormalized by the annulus size of the respective bin $\pi$ (bin2-(bin-1)2). The density histograms were generated per gene and compared to the histogram of all genes within the panel to capture relative overrepresentation.

**Transwell migration assays**
**Isolation of murine T cells.** Spleens were dissociated through 70 μm strainers into cold PBS. Erythrocytes were removed using a gentle hypotonic lysis adapted for Cas9/dCas9 mice (1 mL VE H$_2$O, 7 mL RPMI1640 medium, layered over 1 mL FBS, centrifuged $500 \times g$, 5 min). Cell suspensions were washed, filtered, and centrifuged ($400 \times g$, 5 min, 4 °C). Splenocytes were resuspended in PBS and used for downstream applications.

**CRISPR-based CXCR3 knock-out in murine T-cells.** A PMSC-U6-sgRNA_scaffold-PGK-PuroR-BFP plasmid was used as the backbone for cloning single guide RNAs (sgRNAs) via Golden Gate assembly using BbsI. Three sgRNAs targeting Cxcr3 were designed using the VBC Score tool (https://www.vbc-score.org/), and the leading candidate was chosen for downstream experiments. A non-targeting sgRNA was included as a negative control (sequences provided in Supplementary Information). Retroviral particles were produced by transfecting Plat-E packaging cells with CXCR3-targeting or non-targeting (NT) gRNA constructs together with the helper plasmid pCL-Eco using Lipofectamine 2000 (Thermo Fisher Scientific). Viral supernatants were collected 48 h post-transfection, filtered through 0.45 μm filters, and

used immediately for transduction. CD3$^+$ T cells were isolated from spleens of Rosa26-Cas9 mice using mouse Pan T-cell isolation kit I (Miltenyi Biotec) according the manufacturer protocol. Cells were activated for 24 h in 24-well plates pre-coated with anti-CD3ε (1 μg/ml) and addition of anti-CD28 (2 μg/ml) and 100 U/ml recombinant human IL-2 at a cell density of $3 \times 10^6$ cells/ml. RetroNectin-coated 24-well plates (25 μg/ml, Takara) were preincubated with PBS containing 2% BSA, rinsed, and loaded with viral supernatant. Plates were centrifuged at $2000 \times g$ for 2 h at 32 °C, after which $1 \times 10^6$ activated T cells were added per well and spinoculated at 1500 rpm for 10 min. Cells were cultured for 4 days at 37 °C in IL-2–supplemented lymphocyte medium and split when necessary. Transduction efficiency was assessed by BFP$^+$ and GFP$^+$ fluorescence from the retroviral constructs and Rosa26-Cas9 mouse line, respectively. CXCR3 knockout was confirmed by flow cytometry. NT-transduced cells (NT KO) served as controls in all downstream assays.

**CXCL10-mediated migration and CXCR3 blocking assay.** For transwell migration assays, Millicell® hanging inserts (pore size 5 μm, Merck) were used. Recombinant mouse CXCL10 (200 ng/ml in serum-free RPMI) was added to the basolateral side (lower chamber) in a total volume of 600 μl. $1 \times 10^6$ CXCR3 KO or NT KO T-cells were added in a volume of 200 μl to the apical side (upper insert) in serum-free RPMI1640. Where indicated, a blocking anti-CXCR3 antibody (1 μg/ml, Biolegend, Clone:CXCR3-173) was included in the upper chamber. Cells were incubated for 8 h at 37 °C. Migrated cells in the lower chamber were collected, stained for CD3(PercP Cy5.5, Biolegend 100218), CD8(PE-Cy7, Biolegend 100722), and CXCR3 (APC, Biolegend 126512), and analyzed by flow cytometry (ZE5 Cell analyzer Biorad). eFluor 780 fixable viability dye (Invitrogen) was used according to the manufacturer's protocol to exclude dead cells, and counting beads (123count eBeads, Invitrogen) were used to quantify absolute counts of migrated cells. GFP and BFP fluorescence were used to distinguish Cas9-expressing migrating cells and transduced populations, respectively.

**ISG induction and ISG-T cell mediated migration assays.** CD8$^+$ T cells were isolated from spleens of Rosa26-Cas9$^{-/-}$ mice (GFP-) using mouse CD8a+ T-cell isolation kit (Miltenyi Biotec) for isolating untouched CD8 T-cells according to the manufacturer's instructions. Cells were activated for 24 h in 24-well plates pre-coated with anti-CD3ε (1 μg/ml) and addition of anti-CD28 (2 μg/ml) and 100 U/ml recombinant human IL-2 at a cell density of $3 \times 10^6$ cells/ml. Activated CD8$^+$ T cells were harvested, washed in TexMACS medium (serum-free) to remove residual FBS, and seeded in the basolateral side of transwell plates ($0.5 \times 10^6$ cells per well) in TexMACS medium.

ISG induction was achieved by treating CD8 + T cells in the basolateral well with the murine STING agonist DMXAA (10 μg/ml, Sigma-Aldrich) for 5 h at 37 °C. Control wells received vehicle (TexMACS medium with 0.1% DMSO). Similar to the CXCL10-mediated migration assay, $1 \times 10^6$ transduced CXCR3 KO and NT KO T-cells were added in a volume of 200 μl to the apical side in serum-free RPMI and assay incubated for 8 h at 37 °C. Number of migrated cells was assessed as above.

**Data visualization.** Data visualization was done using GraphPad Prism (v10). Visualization of single-cell data was performed using Seurat (v 4.4.0) with additional customization using ggplot2 (v3.5.1) where required. Graph types and statistical annotations are described in the corresponding figure legends.

**Reporting summary**
Further information on research design is available in the Nature Portfolio Reporting Summary linked to this article.

## Data availability
All single-cell sequencing reads, count matrices, and post-processed files are available at https://www.ncbi.nlm.nih.gov/geo/query/acc.cgi?acc=GSE280018. All spatial transcriptomic raw data and cell segmented data generated in this study are available at https://www.ncbi.nlm.nih.gov/geo/query/acc.cgi?acc=GSE325911. All raw data generated in this study are provided in the Source Data file. Source data are provided with this paper.

## Code availability
Analysis software and code used from publicly available software are described in the manuscript and listed in the reporting summaries. Custom code for downstream spatial transcriptomic analysis is available under: 10.5281/zenodo.19220610. The methodology has been described in detail in the Methods section of the manuscript.

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

## Acknowledgements

We acknowledge the support of the DKFZ Light Microscopy Facility, Genomics and Proteomics Core Facility, Center for Preclinical Research, Flow Cytometry Core Facility, and single-cell Open Lab. We thank Dr. Franziska Blaeschke and Konstantin Loher for providing plasmids for CRISPR-based KO experiments. We acknowledge the data storage service SDS@hd supported by the Ministry of Science, Research, and the Arts Baden-Württemberg (MWK) and the German Research Foundation (DFG) through grant INST 35/1314-1 FUGG and INST 35/1503-1 FUGG. We acknowledge the Department of Biomedical Informatics, Mannheim Institute for intelligent Systems in Medicine, Medical Faculty Mannheim, Heidelberg University, Mannheim, Germany for the management of our RedCap database. This study was supported by grants from the Dr. Rolf M. Schwiete Foundation, the Swiss Cancer Foundation (Swiss Bridge Award), the Else Kröner Fresenius Foundation (2019_EKMS.49), the University Heidelberg Foundation (Hella Bühler Award), the DFG (German Research Foundation), project 404521405 (SFB1389 UNITE Glioblastoma B01 and B03), the Hertie Foundation, and the University of Heidelberg, ExploreTech! Grant to L.B. The DFG, project 394046768 (SFB1366/2 C01), the DFG RTG2727 InCheck project 445549683 PL-315/9-1, and ERC grant CENTRIC-BRAIN (grant agreement no. 101141901) to M.P. The DFG Priority Program "Local and Peripheral Drivers of Microglial Diversity and Function" (SPP 2395) project PL315/10-1 and PL315/10-2 to M.P. and L.B., the Hertie Foundation (Seed Call TIRE-CAA) to K.S., K.D., and L.B. J.M. was supported by the Mildred-Scheel fellowship by the German Cancer Aid. C.T. was supported by the European Training Network GLIORESOLVE (grant agreement no. 101073386). This work is supported by the Health + Life Science Alliance Heidelberg Mannheim and received state funds approved by the State Parliament of Baden-Württemberg.

## Author contributions

J.M. designed and performed experiments, analyzed and interpreted data, and wrote the manuscript. K.S. conceptualized the study, designed and performed experiments, analyzed and interpreted data, and wrote the manuscript. J.R. analyzed and interpreted data. L.H., C.T.D., S.G., V.S., and K.J. performed experiments. K.D. and L.F. interpreted data and edited the manuscript. J.H. provided tissue, interpreted data, and edited the manuscript, M.F. provided resources, M.P., co-conceptualized the study, provided resources, interpreted data and edited the manuscript. L.B. conceptualized the study, interpreted data, and wrote the manuscript.

## Funding

## Competing interests

M.P.s is the founder of TcellTech. All other authors declare no conflict.
