## [Transparent Peer Review file · Nature Communications]

Type I Interferon drives T cell responses to amyloid beta in the central nervous system

Corresponding Author: Professor Lukas Bunse

Version 0:

Reviewer comments:

Reviewer #1

(Remarks to the Author)

In this study, Michel et al. investigated immune microenvironment dynamics in APP23 transgenic mice using single-cell transcriptomics. They report a significant increase in T-cell populations, particularly CD8+ T-cells, during late stages of disease progression. Further, the authors identify an A β plaque-associated subset of CD8+ T-cells expressing interferon-stimulated genes.

While some of the findings presented are intriguing and contribute to our understanding of T-cell dynamics in neurodegenerative disease, I found several conclusions to be supported by limited evidence. This, in my view, precludes publication of the manuscript in its current form. Specifically, some claims - such as the role of CXCL10 in driving T-cell infiltration and its link to activation-induced exhaustion - require additional experimental validation. Furthermore, the methodological details and data presentation in certain areas could be improved to enhance clarity. Below, I provide detailed comments and suggestions to address these concerns.

1)

The statement, "Thus, APP23 mice provide a model that resembles progressive amyloidosis and cognitive impairment where the onset and development of disease often manifest in old age," may overstate the observations by the authors. While the observed decline in nesting ability is an important result, it alone does not provide sufficient evidence to conclusively demonstrate cognitive impairment. Nesting behavior could reflect other neurological deficits, such as motor dysfunction, changes in motivation, or anxiety, which are not directly indicative of cognitive decline.

Additional cognitive tests would be needed to substantiate the claim that APP23 mice specifically model cognitive decline alongside amyloidosis. Alternatively, the authors could strengthen their argument by citing previous studies that have demonstrated cognitive deficits in this model.

2)

"Next, to understand the changes in the immune microenvironment with progressive disease, we subjected CD45+ cells, isolated from the brains of early and late mice and corresponding wild-type littermates (Wt) to single-cell RNA (scRNA) and VDJ (scVDJ) sequencing (Fig 1a, Fig S2a)."

It is unclear to me whether the animals were perfused prior to tissue collection for the single-cell RNA-seq experiment. Since I would expect that perfusion could significantly affect the cellular composition, particularly by removing peripheral immune cells, I think it would be helpful for the reader if the authors could clarify this point. This information would help in interpreting what the observed cell type composition represents. If the animals were not perfused, it would be valuable for the authors to explain their rationale. Explicitly addressing this would enhance the study's transparency and provide important context for readers evaluating the immune microenvironment data.

3)

In Figure 1c, the authors present the frequency of CD3+ T-cells, CD4+ T-cells, and CD8+ T-cells within all immune cells, with the y-axis labeled as "% of CD45+ cells." In the Tg late group, the values for CD3+ T-cells appear to range between 0.1 and 0.15. It is unclear whether these values should be interpreted as 10-15% or 0.1-0.15%. I suggest the authors clarify the y-axis labeling and explicitly state how the values should be interpreted to avoid any ambiguity.

4)

Figure 1f: "Ingenuity pathway analysis (IPA) comparing early APP23-tg with wild-type mice, late APP23-tg with wild-type mice and early with late APP23-tg mice."

The authors explain in the methods section that they performed a differential expression analysis before conducting the Ingenuity Pathway Analysis (IPA). However, it remains unclear which cells were included in this analysis. Was the analysis performed on all CD45+ cells or on a specific subset?

5)

The authors report a gradual increase in T-cell abundance with age, corresponding to increasing plaque burden, peaking at 20 months (Fig. 2b,c). However, in the immunohistochemistry images presented in Figure 2a, the T-cells are not clearly identifiable. To enhance clarity, I recommend annotating these images with arrows or markers to indicate what the authors interpret as T-cells.

Additionally, in Figure 2a, what appears to be amyloid plaques seems to exhibit more blue (DAPI) than green (amyloid beta) staining. This observation is unexpected to me and makes it challenging to interpret the images. I suggest the authors elaborate on their interpretation of the IHC images, potentially annotating regions they identify as plaques. Providing more detail on how the images support the quantitative data would strengthen the overall presentation.

6)

If I interpret Figure 1c correctly, CD3+ T cells constitute approximately 10–15% of all CD45+ cells. However, in Figure 2b, the authors report T-cell abundance based on immunohistochemistry as T cells per total cells, with values at later time points (20 and 24 months) also ranging between 0.1 and 0.15 (i.e., 10-15%). This suggests a potential discrepancy: if T cells make up 10–15% of CD45+ cells, I would expect their percentage relative to all cells to be much lower.

I recommend that the authors clarify this apparent discrepancy or explain if my interpretation is incorrect.

7)

For the single-cell spatial transcriptomics analysis, I believe the authors could provide additional methodological details to help readers better understand their approach. Specifically, it would be helpful to include information on how many animals were analyzed, how many sections per animal were used, and the specific area that was examined.

Additionally, I suggest the authors provide a more basic analysis of the cell types identified through spatial transcriptomics. For example, how does the expression of marker genes look across all the identified cell types? The authors could include violin plots to illustrate the distribution of marker gene expression. Furthermore, it would be particularly informative to assess whether the cluster with higher T-cell scores, shown in Figure S6b, is significantly more abundant in APP23 mouse brains compared to control brains.

8)

The authors conclude that "T-cell infiltration during neuroinflammation is driven by activation of CXCR3 by CXCL10." However, the evidence presented does not sufficiently support this conclusion. To substantiate this claim, additional experiments would be necessary, such as functional studies demonstrating a direct role of CXCR3-CXCL10 signaling in T-cell infiltration. I encourage the authors to either provide further experimental evidence or revise the conclusion to better align with the data presented.

9)

The authors state that "As PDCD1 is mostly expressed by exhausted T-cells this observation corroborates our preclinical finding that CXCL10-driven chemotaxis leads to activation-induced exhaustion verging on plaques in CNS amyloid pathology." However, I am concerned that the evidence provided does not sufficiently support this claim.

Firstly, as mentioned above, the authors provide limited evidence demonstrating that the observed chemotaxis is specifically induced by CXCL10. Secondly, I do not see clear evidence linking chemotaxis to activation-induced exhaustion in the presented data. To substantiate these claims, the authors would need to provide additional experimental support.

Minor comments:

Figure S4a: The different conditions are not labeled, and there also appears to be no color legend.

In Figure 3f, the dot plot on the right side of the panel appears to show four distinct groups separated along the y-axis. However, the labels for these groups seem to be missing.

In Figure 6c what does P1, P2, P3, P4 stand for?

Reviewer #2

(Remarks to the Author)

In this manuscript, Julius Michel et al report that 1) CD8+ T cell accumulation increases in APP23-tg mice. 2) Those T cells cluster around the amyloid plaque. 3) CD8+ T cells express interferon-stimulated genes (ISGs). 4) ISG CD8+ T cells recruit other T cells infiltration via CXCL10-CXCR3. The research and findings are interesting. There are, however, major and minor concerns that need to be addressed.

1. Please show CD3+, CD4+ and CD8+ T cell immunostaining in the brain of APP23 mice in Fig 2. It is important to know

which population of T cells are localized around amyloid plaque.

2. ISGs are expressed in some populations of microglia and CD8+ T cells, especially at the late stage. Please list the evidence that ISG+ CD8+ T cells recruit more T cells infiltrating the brain, in addition to the cell-cell interaction informatic analysis. The authors should consider 1) CXCL10 neutralize antibody to block the CXCL10-CXCR3 signaling 2) CD8-cre; CXCL10; APP23 mice model 3) CXCL10 and CXCR3 co-staining with microglia and CD8+ T cells in mice and human tissue.

3. The CD8+ T cell clonal results (Fig.5) are not convincing. As the authors have TCR information. Please show 1) TCR clonal distribution and clonal type in T cells in different stages and groups. 2) Please show the top clones in the proliferating T cells and the predicted peptides.

4. What's the function of IL17+ T cells and NKT cells in the brain of late APP23 mice? Please show the differential expressed genes and gene function analysis in APP23 late vs. APP23 early and APP23 late vs. WT late.

Reviewer #3

(Remarks to the Author)

Julius Michel et al. present a very interesting and methodologically high end bioinformatic and descriptive study on the identity of CD8 T-cells associated with Abeta plaques in an transgenic mouse model of amyloidogenic Alzheimer's disease and in human postmortem AD brain specimen.

The authors confirmed existing data that CD8 T-cells invade the AD brain not in the initial phase of pathology, but at later stages. They also confirm that these cells predimantly localize at amyloid plaques. The statial transcriptomcs data nicely illustrate that in particular the plaque associated CD8 T-cells are characterized by a Type 1 interferon response gene expression profile. This information certainly adds to the existing knowledge, however it has to be stated that in bulk transcriptomic profiling of CD8 T-cells in the context of amyloid pathology interferon signaling has recently been already shown (Altendorfer et al., J Immunol. 2022 Oct 1;209(7):1272-1285. doi: 10.4049/jimmunol.2100737). Finally, authors confirm a potential role of the CXCL10-CXCR3 axis in the infiltration of the CD8 T-cells into the amyloidogenic brain (Jorfi et a., Nat Neurosci. 2023 Sep;26(9):1489-1504.doi: 10.1038/s41593-023-01415-3).

The strength of this manuscript is certainly the high end bioinformatic analysis and the temporal - spatial aspect of the analysis and data. This certainly support the existing ideas on the identity of the CD8 T-cells in the context of AD.

The weakness, however, is the relatively low amount of novelty. The data are rather supportive for the current perspective in the identity of the CD8 T-cells in AD, but don't add substantial novelty. Most importantly, the work is missing any functional relevance, it stays rather descriptive (although at a high level of bioinformatics).

Therefore, I am extremely supportive for this work to be published, but not in an high impact journal such as Nature Communications. In case that authors could provide more functional data / relevance, I would be in favour of seeing it published in Nature Communications, or even higher.

Reviewer #4

(Remarks to the Author)

The paper is well-written, and the experimental design is solid. The authors identified that and increased in T-cell populations in the AD mouse model and studied its spatial profile using spatial transcriptomics. The novelty of the paper may be somewhat diminished as some claims have been previously reported. However, the findings may still be acceptable for publication as it utilized a new technological and revealed more details. However, I still have major concerns regarding to the data quality, analysis and part of their interpretation that needs to be addressed before publication.

Specific Comments:

1. Fig. 2a. The figure is somewhat unclear. The authors should specify the anatomical region represented. The cell density revealed by DAPI staining appears very low—does this reflect a specific region or an ultra-thin section? The images in Fig. 2e seem more consistent.
2. Fig. 2f. The figure suggests that over 70% of the cells are T-cells. Is this correct? The 200 μ m scale bar indicates that a 1x1 mm² area is predominantly occupied by T-cells, which is remarkable. Is this observation general across different anatomical regions or specific to the chosen area? The legend or results section does not specify this. Could the authors quantify T-cell density across different regions? The high percentage warrants further clarification.
3. Fig. 1c vs. Fig. 2b. In Fig. 1c, after FACS, CD3+ T-cells constitute about 0.1–0.2% of the CD45+ T-cells (likely mislabeled; it should be 10–20%, please confirm). In Fig. 2b, which does not involve FACS, CD3+ cells still make up about 10–15% of all cells. Could the authors explain this significant discrepancy?
4. Fig. 2g. The authors claim that “this data suggests that although the T-cell response is directed towards A β , infiltrating T-cells have a greater attraction towards parenchymal plaques compared to vascular amyloid deposits.” The data only suggest a higher correlation, not necessarily stronger attraction. Rather, the data imply that, per 10 vascular amyloid deposits, T-cells/total cells = 0.1–0.15, whereas for 10 parenchymal plaques, T-cells/total cells = 0.05. The interpretation should be reconsidered.
5. Spatial Transcriptomics. The description of spatial transcriptomics needs further details. The Molecular Cartography

technique is not cited, and its methodology is unclear. The authors should explain how the technique works in principle for readers. For instance, is it imaging-based or sequencing-based? Are barcoded probes used, and what is the expected error rate? Additional information about the distribution of transcripts per segmented cell and the correlation with scRNA-seq data would be helpful. An illustration similar to Fig. S5 would improve clarity.

6. Fig. 3j. How is distance 0 determined? Is it the center of the plaque or the boundary of the closest plaque? If it is the center, Fig. 3g shows this area as devoid of T-cells. If it is the boundary, how is the exact point determined for each gene? Were different transcripts assigned different references? The overall trend looks reasonable, but this should be clarified.

7. Figs. 3g and 4. Which anatomical region do the samples come from? As the authors note regional specificity in changes (e.g., cortex, hippocampus, and subcortical areas), this information is necessary. In Fig. 3g, is the image raw, processed, or superimposed? Were the A β antibodies consistent with those in previous staining? Could the authors differentiate parenchymal and vascular amyloid plaques, as mentioned in Fig. 2? Does the spatial transcriptomics data reflect similar differences?

8. Fig. S6. Can the authors subcluster T-cells using spatial transcriptomics data? In Fig. S6c, the neuronal population percentage is very low. Is the data quality sufficient for the proposed study? How consistent is the cell composition across different animals, and how does it compare with scRNA-seq data?

9. Fig. 6. Results analogous to those in Figs. 3–4 should be presented. Specifically, the authors should include dot plots for transcripts, their spatial proximity to plaques, and quality evaluations of the human spatial transcriptomics data. UMAP plots, clustering, and cell annotations (as in Fig. S6) would provide a better basis for assessment.

10. Methods. In the IPA section, it states, "To obtain heatmaps of 'Diseases and functions' and 'xx'." Is "xx" included intentionally, or is it a mistake?

Version 1:

Reviewer comments:

Reviewer #1

(Remarks to the Author)

The authors have adequately addressed most of my previous comments. I still have several questions related to my initial comment number 7 regarding the single-cell spatial transcriptomics analysis.

In Figure S6b, the authors annotate a relatively large cluster as T cells. In contrast, the proportions of T cells reported across groups in Figure S6e appear quite low: in Tg late the proportion is approximately 0.01, and in the other groups it is around 0.002–0.003. Visually, however, the cluster annotated as T cells in Figure S6b appears larger than 1% of all cells. The authors should verify whether the reported proportions are accurate and clarify any potential discrepancy between the UMAP visualization and the quantitative estimates.

The authors also state:

"Similarly, cell-segmentation-based analyses confirmed an ISG-upregulating T-cell subset. When compared to other cells, the abundance of cells within this subset was higher in the plaque neighborhood (<69 μ m) (Fig S6a-f)."

Given this claim, I think it would be important to show the expression of canonical T-cell markers within this ISG-upregulating subset. This would help confirm that these cells are indeed T cells rather than another ISG-high population.

Reviewer #3

(Remarks to the Author)

Congratulations to the authors. The revised version shows an immense improvement and an overwhelming amount of additional data now clearly supporting the authors' conclusion.

All aspects and comments raised on the initial submission are adequately addressed.

Reviewer #4

(Remarks to the Author)

The author addressed most of my concerns.

As the authors propose a conceptual shift from microglia to T cells, and given that spatial transcriptomic data are available, it would be highly informative to leverage these data more fully. Specifically, the authors could examine age-dependent changes in the spatial distribution and density (or cellular composition) of different immune cell types as a function of distance from amyloid plaques. An analysis analogous to the stacked cell-composition plot in Figure 1, but incorporating spatial proximity to plaques, would be particularly compelling. Without such analyses, the spatial transcriptomic data are underutilized and do not substantially extend the insights already provided by the scRNA-seq results.

Version 2:

Reviewer comments:

Reviewer #1

(Remarks to the Author)

The authors have addressed my comments satisfactorily.

Reviewer #4

(Remarks to the Author)

The authors have addressed my previous comments, and I have no further concerns. Congratulations!

Rebuttal

“Type-I Interferon drives T-cell responses to Amyloid-beta in the central nervous system”

REVIEWER COMMENTS

Reviewer #1 (Remarks to the Author):

In this study, Michel et al. investigated immune microenvironment dynamics in APP23 transgenic mice using single-cell transcriptomics. They report a significant increase in T-cell populations, particularly CD8+ T-cells, during late stages of disease progression. Further, the authors identify an A β plaque-associated subset of CD8+ T-cells expressing interferon-stimulated genes. While some of the findings presented are intriguing and contribute to our understanding of T-cell dynamics in neurodegenerative disease, I found several conclusions to be supported by limited evidence. This, in my view, precludes publication of the manuscript in its current form. Specifically, some claims - such as the role of CXCL10 in driving T-cell infiltration and its link to activation-induced exhaustion - require additional experimental validation. Furthermore, the methodological details and data presentation in certain areas could be improved to enhance clarity. Below, I provide detailed comments and suggestions to address these concerns.

1)

The statement, "Thus, APP23 mice provide a model that resembles progressive amyloidosis and cognitive impairment where the onset and development of disease often manifest in old age," may overstate the observations by the authors. While the observed decline in nesting ability is an important result, it alone does not provide sufficient evidence to conclusively demonstrate cognitive impairment. Nesting behavior could reflect other neurological deficits, such as motor dysfunction, changes in motivation, or anxiety, which are not directly indicative of cognitive decline. Additional cognitive tests would be needed to substantiate the claim that APP23 mice specifically model cognitive decline alongside amyloidosis. Alternatively, the authors could strengthen their argument by citing previous studies that have demonstrated cognitive deficits in this model.

2)

“Next, to understand the changes in the immune microenvironment with progressive disease, we subjected CD45+ cells, isolated from the brains of early and late mice and corresponding wild-type littermates (Wt) to single-cell RNA (scRNA) and VDJ (scVDJ) sequencing (Fig 1a, Fig S2a).”

It is unclear to me whether the animals were perfused prior to tissue collection for the single-cell RNA-seq experiment. Since I would expect that perfusion could significantly affect the cellular composition, particularly by removing peripheral immune cells, I think it would be helpful for the reader if the authors could clarify this point. This information would help in interpreting what the observed cell type composition represents. If the animals were not perfused, it would be valuable for the authors to explain their rationale. Explicitly addressing this would enhance the study's transparency and provide important context for readers evaluating the immune microenvironment data.

3)

In Figure 1c, the authors present the frequency of CD3+ T-cells, CD4+ T-cells, and CD8+ T-cells within all immune cells, with the y-axis labeled as "% of CD45+ cells." In the Tg late group, the values for CD3+ T-cells appear to range between 0.1 and 0.15. It is unclear whether these values should be interpreted as 10-15% or 0.1-0.15%. I suggest the authors clarify the y-axis labeling and explicitly state how the values should be interpreted to avoid any ambiguity.

4)

Figure 1f: “Ingenuity pathway analysis (IPA) comparing early APP23-tg with wild-type mice, late APP23-tg with wild-type mice and early with late APP23-tg mice.”

The authors explain in the methods section that they performed a differential expression analysis before conducting the Ingenuity Pathway Analysis (IPA). However, it remains unclear which cells were included in this analysis. Was the analysis performed on all CD45+ cells or on a specific subset?

5)

The authors report a gradual increase in T-cell abundance with age, corresponding to increasing plaque burden, peaking at 20 months (Fig. 2b,c). However, in the immunohistochemistry images presented in Figure 2a, the T-cells are not clearly identifiable. To enhance clarity, I recommend annotating these images with arrows or markers to indicate what the authors interpret as T-cells.

Additionally, in Figure 2a, what appears to be amyloid plaques seems to exhibit more blue (DAPI) than green (amyloid beta) staining. This observation is unexpected to me and makes it challenging to interpret the images. I suggest the authors elaborate on their interpretation of the IHC images, potentially annotating regions they identify as plaques. Providing more detail on how the images support the quantitative data would strengthen the overall presentation.

6)

If I interpret Figure 1c correctly, CD3+ T cells constitute approximately 10–15% of all CD45+ cells. However, in Figure 2b, the authors report T-cell abundance based on immunohistochemistry as T cells per total cells, with values at later time points (20 and 24 months) also ranging between 0.1 and 0.15 (i.e., 10-15%). This suggests a potential discrepancy: if T cells make up 10–15% of CD45+ cells, I would expect their percentage relative to all cells to be much lower.

I recommend that the authors clarify this apparent discrepancy or explain if my interpretation is incorrect.

7)

For the single-cell spatial transcriptomics analysis, I believe the authors could provide additional methodological details to help readers better understand their approach. Specifically, it would be helpful to include information on how many animals were analyzed, how many sections per animal were used, and the specific area that was examined.

Additionally, I suggest the authors provide a more basic analysis of the cell types identified through spatial transcriptomics. For example, how does the expression of marker genes look across all the identified cell types? The authors could include violin plots to illustrate the distribution of marker gene expression. Furthermore, it would be particularly informative to assess whether the cluster with higher T-cell scores, shown in Figure S6b, is significantly more abundant in APP23 mouse brains compared to control brains.

8)

The authors conclude that "T-cell infiltration during neuroinflammation is driven by activation of CXCR3 by CXCL10." However, the evidence presented does not sufficiently support this conclusion. To substantiate this claim, additional experiments would be necessary, such as functional studies demonstrating a direct role of CXCR3-CXCL10 signaling in T-cell infiltration. I encourage the authors to either provide further experimental evidence or revise the conclusion to better align with the data presented.

9)

The authors state that "As PDCD1 is mostly expressed by exhausted T-cells this observation corroborates our preclinical finding that CXCL10-driven chemotaxis leads to activation-induced exhaustion verging on plaques in CNS amyloid pathology." However, I am concerned that the evidence provided does not sufficiently support this claim.

Firstly, as mentioned above, the authors provide limited evidence demonstrating that the observed chemotaxis is specifically induced by CXCL10. Secondly, I do not see clear evidence linking chemotaxis to activation-induced exhaustion in the presented data. To substantiate these claims, the authors would need to provide additional experimental support.

Minor comments:

Figure S4a: The different conditions are not labeled, and there also appears to be no color legend.

In Figure 3f, the dot plot on the right side of the panel appears to show four distinct groups separated along the y-axis. However, the labels for these groups seem to be missing.

In Figure 6c what does P1, P2, P3, P4 stand for?

Reviewer #2 (Remarks to the Author):

In this manuscript, Julius Michel et al report that 1) CD8+ T cell accumulation increases in APP23-tg mice. 2) Those T cells cluster around the amyloid plaque. 3) CD8+ T cells express interferon-stimulated genes (ISGs). 4) ISG CD8+ T cells recruit other T cells infiltration via CXCL10-CXCR3. The research and findings are interesting. There are, however, major and minor concerns that need to be addressed.

1. Please show CD3+, CD4+ and CD8+ T cell immunostaining in the brain of APP23 mice in Fig 2. It is important to know which population of T cells are localized around amyloid plaque.

2. ISGs are expressed in some populations of microglia and CD8+ T cells, especially at the late stage. Please list the evidence that ISG+ CD8+ T cells recruit more T cells infiltrating the brain, in addition to the cell-cell interaction informatic analysis. The authors should consider 1) CXCL10 neutralize antibody to block the CXCL10-CXCR3 signaling 2) CD8-cre; CXCL10; APP23 mice model 3) CXCL10 and CXCR3 co-staining with microglia and CD8+ T cells in mice and human tissue.

3. The CD8+ T cell clonal results (Fig.5) are not convincing. As the authors have TCR information. Please show 1) TCR clonal distribution and clonal type in T cells in different stages and groups. 2) Please show the top clones in the proliferating T cells and the predicted peptides.

4. What's the function of IL17+ T cells and NKT cells in the brain of late APP23 mice? Please show the differential expressed genes and gene function analysis in APP23 late vs. APP23 early and APP23 late vs. WT late.

Reviewer #3 (Remarks to the Author):

Julius Michel et al. present a very interesting and methodologically high end bioinformatic and descriptive study on the identity of CD8 T-cells associated with Abeta plaques in an transgenic mouse model of amyloidogenic Alzheimer's disease and in human postmortem AD brain specimen.

The authors confirmed existing data that CD8 T-cells invade the AD brain not in the initial phase of pathology, but at later stages. They also confirm that these cells predimantly localize at amyloid plaques. The spatial transcriptomics data nicely illustrate that in particular the plaque associated CD8 T-cells are characterized by a Type 1 interferon response gene expression profile. This information certainly adds to the existing knowledge, however it has to be stated that in bulk transcriptomic profiling of CD8 T-cells in the context of amyloid pathology interferon signaling has recently been already shown (Altendorfer et al., J Immunol. 2022 Oct 1;209(7):1272-1285. doi: 10.4049/jimmunol.2100737). Finally, authors confirm a potential role of the CXCL10-CXCR3 axis in the infiltration of the CD8 T-cells into the amyloidogenic brain (Jorfi et al., Nat Neurosci. 2023 Sep;26(9):1489-1504. doi: 10.1038/s41593-023-01415-3).

The strength of this manuscript is certainly the high end bioinformatic analysis and the temporal - spatial aspect of the analysis and data. This certainly support the existing ideas on the identity of the CD8 T-cells in the context of AD.

The weakness, however, is the relatively low amount of novelty. The data are rather supportive for the current perspective in the identity of the CD8 T-cells in AD, but don't add substantial novelty. Most importantly, the work is missing any functional relevance, it stays rather descriptive (although at a high level of bioinformatics).

Therefore, I am extremely supportive for this work to be published, but not in an high impact journal such as Nature Communications. In case that authors could provide more functional data / relevance, I would be in favour of seeing it published in Nature Communications, or even higher.

Reviewer #4 (Remarks to the Author):

The paper is well-written, and the experimental design is solid. The authors identified that and increased in T-cell populations in the AD mouse model and studied its spatial profile using spatial transcriptomics. The novelty of the paper may be somewhat diminished as some claims have been previously reported. However, the findings may still be acceptable for publication as it utilized a new technological and revealed more details. However, I still have major concerns regarding to the data quality, analysis and part of their interpretation that needs to be addressed before publication.

Specific Comments:

1. Fig. 2a. The figure is somewhat unclear. The authors should specify the anatomical region represented. The cell density revealed by DAPI staining appears very low—does this reflect a specific region or an ultra-thin section? The images in Fig. 2e seem more consistent.

2. Fig. 2f. The figure suggests that over 70% of the cells are T-cells. Is this correct? The 200 μm scale bar indicates that a $1 \times 1 \text{ mm}^2$ area is predominantly occupied by T-cells, which is remarkable. Is this observation general across different anatomical regions or specific to the chosen area? The legend or results section does not specify this. Could the authors quantify T-cell density across different regions? The high percentage warrants further clarification.

3. Fig. 1c vs. Fig. 2b. In Fig. 1c, after FACS, CD3+ T-cells constitute about 0.1–0.2% of the CD45+ T-cells (likely mislabeled; it should be 10–20%, please confirm). In Fig. 2b, which does not involve FACS, CD3+ cells still make up about 10–15% of all cells. Could the authors explain this significant discrepancy?

4. Fig. 2g. The authors claim that “this data suggests that although the T-cell response is directed towards $A\beta$, infiltrating T-cells have a greater attraction towards parenchymal plaques compared to vascular amyloid deposits.” The data only suggest a higher correlation, not necessarily stronger attraction. Rather, the data imply that, per 10 vascular amyloid deposits, T-cells/total cells = 0.1–0.15, whereas for 10 parenchymal plaques, T-cells/total cells = 0.05. The interpretation should be reconsidered.

5. Spatial Transcriptomics. The description of spatial transcriptomics needs further details. The Molecular Cartography technique is not cited, and its methodology is unclear. The authors should explain how the technique works in principle for readers. For instance, is it imaging-based or sequencing-based? Are barcoded probes used, and what is the expected error rate? Additional information about the distribution of transcripts per segmented cell and the correlation with scRNA-seq data would be helpful. An illustration similar to Fig. S5 would improve clarity.

6. Fig. 3j. How is distance 0 determined? Is it the center of the plaque or the boundary of the closest plaque? If it is the center, Fig. 3g shows this area as devoid of T-cells. If it is the boundary, how is the exact point determined for each gene? Were different transcripts assigned different references? The overall trend looks reasonable, but this should be clarified.

7. Figs. 3g and 4. Which anatomical region do the samples come from? As the authors note regional specificity in changes (e.g., cortex, hippocampus, and subcortical areas), this information is necessary. In Fig. 3g, is the image raw, processed, or superimposed? Were the $A\beta$ antibodies consistent with those in previous staining?

Could the authors differentiate parenchymal and vascular amyloid plaques, as mentioned in Fig. 2? Does the spatial transcriptomics data reflect similar differences?

8. Fig. S6. Can the authors subcluster T-cells using spatial transcriptomics data? In Fig. S6c, the neuronal population percentage is very low. Is the data quality sufficient for the proposed study? How consistent is the cell composition across different animals, and how does it compare with scRNA-seq data?

9. Fig. 6. Results analogous to those in Figs. 3–4 should be presented. Specifically, the authors should include dot plots for transcripts, their spatial proximity to plaques, and quality evaluations of the human spatial transcriptomics data. UMAP plots, clustering, and cell annotations (as in Fig. S6) would provide a better basis for assessment.

10. Methods. In the IPA section, it states, “To obtain heatmaps of ‘Diseases and functions’ and ‘xx’.” Is “xx” included intentionally, or is it a mistake?

Response to Reviewer #1

1)

We agree that the decline in nesting ability alone does not conclusively demonstrate cognitive impairment and have therefore toned down the respective statement in the manuscript. Our intention was to describe the APP23 model as one that recapitulates progressive amyloidosis and associated neurological alterations with aging. We also acknowledge that previous studies have reported age-related cognitive changes in APP23 mice (Kelly et al., 2003; Van Dam et al., 2003) and have cited these studies for reference. The revised text now reads:

“APP23-tg mice have been shown to demonstrate age-dependent alterations using the Morris Water Maze test (Kelly u. a., 2003; Van Dam u. a., 2003). (.....) Thus, APP23 mice provide a model that resembles progressive amyloidosis and associated age-dependent neurological alterations”

2)

In our study, the animals were not perfused prior to brain tissue collection. We acknowledge that this decision may result in the inclusion of peripheral immune cells, which could influence the overall composition of immune cell populations observed in the single-cell RNA-seq data. However, we chose this approach to ensure that we captured a comprehensive representation of the immune microenvironment, including both resident brain and infiltrating immune cells. This is particularly relevant given the progressive nature of amyloidosis in the APP23 mice, where peripheral immune cell infiltration could be influenced by local cues as we have then subsequently demonstrated. We have now updated the corresponding Methods section (*“Isolation of brain-associated immune cells”*) in the manuscript to explicitly include the information on perfusion and our rationale.

“brains were isolated without prior perfusion to capture a comprehensive representation of the immune microenvironment, including both resident brain and infiltrating immune cells.....”

3)

This was a labeling error and we thank the reviewer for pointing it out. These values are in percentage, and we have corrected the axis labels in the corresponding graph (**Fig 1c**).

Fig 1c:

4)

The Ingenuity Pathway Analysis (IPA) was performed on differentially expressed genes identified from **all CD45+ cells** rather than a specific immune cell subset. Our goal was to capture broad immune-related transcriptional changes occurring in the brain during disease progression. To ensure clarity, we have revised the corresponding Methods section to explicitly state that differential expression analysis was conducted on all CD45+ cells prior to performing IPA (“*Single-cell transcriptomic analyses. Sub-section: Ingenuity pathway analyses*”). Additionally, we have updated the figure legend for **Fig 1f** to reinforce this point.

Methods section: “*For all CD45+ cells differentially expressed (DE) genes between conditions (APP23tg versus Wt) or time points (late versus early) were identified.....*”

Figure legend: “*(f) Ingenuity pathway analysis (IPA) depicting the comparison of CD45+ cells between early APP23-tg and wild-type mice, late APP23-tg and wild-type mice and early and late APP23-tg mice.....*”

5)

To address the reviewer’s concern regarding the visibility of T cells in the immunofluorescence images, we have revised **Fig 2a** to improve clarity and visual confirmation of T cell localization. Specifically, we now provide images at higher magnification and have added arrows to clearly indicate the cells identified as T cells. In addition, we include the corresponding tile scans showing the precise locations of the magnified regions, which are now presented in **Fig S5b** for reference. These

additions allow for better visualization and verification of the T cells within the tissue architecture. The corresponding figure legend has also been updated accordingly to reflect these changes.

Fig 2a:

Figure legend of Fig 2a: “...T cells are indicated by arrows...”

Fig S5b:

With regards to the appearance of amyloid plaques, prominent staining with DAPI is due to the strong autofluorescence of amyloid deposits in the blue channel, which can sometimes appear more intense than the Aβ-specific staining, particularly in aged tissue with dense plaques. Similar observations have been made in previous reports such as by Fu et.al, 2024 (doi: [10.1007/s12264-023-01175-x](https://doi.org/10.1007/s12264-023-01175-x)). We have now included

an explanation in the figure legend of **Fig 2a** to clarify this observation and ensure proper interpretation. “Signal in the DAPI channel from plaques.”

Figure legend: “Signal in the DAPI channel from plaques is a result from expected strong intrinsic blue fluorescence of the cores of plaques (Fu et al., 2024).”

We believe these modifications enhance the clarity of **Fig 2a** and better align the qualitative imaging data with the quantitative findings.

6)

We acknowledge this discrepancy and resolve the latter with additional data. As we were interested in visualizing and quantifying A β plaque associated T cells, we focused on A β -dense regions for analysis of immunofluorescence (IF) stainings. This resulted in an over-enumeration of T cells relative to total cells as these regions are highly abundant in T cells. To compensate for this, we now re-analyzed plaque-devoid regions in sections from the same animals and show that these regions are also scarce in T cells (**Fig S5c**). Taken together, the quantification of CD3+ T cells evens-out relative to all cells in the IF stainings and resolves the discrepancy between the scRNAseq and IF data. This data further quantitatively corroborates that infiltrating T cells in the amyloidogenic brain are mostly found in A β dense regions, as we had already shown qualitatively in **Fig 2f**.

Fig S5c:

7)

To enhance transparency, we have added methodological details specifying the number of animals analyzed, the number of sections per animal, and the specific brain regions examined. Specifically, 2 mice per group were assayed using one section each. Coronal sections of one entire hemisphere were used and ROIs for the transcriptomic readout were marked in the cerebral cortex. These ROIs were determined based on 2 criteria:

1. Areas of the section that were largely intact
2. Areas rich in A β plaques (as determined by immunohistology staining of an adjacent coronal section)

This information is now included in the corresponding Methods section (“*Targeted spatial transcriptomics. Sub-section: Sample preparation*”). We additionally included a figure panel in **Fig S6a**, showing exemplary images of sections with ROIs used for the readout marked.

Methods section: “*Coronal sections of the cerebral cortex of in total eight cryo-preserved APP23-tg and Wt mouse brains (n = 2 per group) were subjected to targeted spatial transcriptomics using the Molecular Cartography™ platform (Resolve Biosciences). For each mouse, a single brain hemisphere was analyzed. A coronally adjacent section was stained with pFTAA dye to detect A β deposits and guide ROI selection within the cerebral cortex for transcript detection. The list of transcripts probed is provided in Supplementary Table 1.*”

Fig S6a:

a

Regarding the request for a more detailed analysis of the identified cell types, we acknowledge that our study employed a targeted spatial transcriptomics approach, which focuses on predefined gene panels rather than an unbiased transcriptome-wide analysis. There were 2 main reasons to pursue this approach:

1. We were primarily interested in querying whether the observations we made using scRNAseq data are brain ‘global’ or spatially restricted, as this has significant consequences on the understanding of how amyloid beta influences T-cell response.
2. For our purposes, single-cell level resolution was crucial which was not achievable through unbiased transcriptome-wide spatial RNA-seq technologies available during the time these experiments were conducted. Furthermore, for the purpose of answering the questions relevant to this study, targeted spatial transcriptomics was sufficient.

In response to the request by the reviewer for additional basic analyses of the data, we have now revised **Fig S6** to reflect the requested information. We have provided an updated UMAP of the clustering with cell-type annotations (**Fig S6b**) and a dot-plot showing expression of canonical cell-specific transcripts between different annotated cell-types (**Fig S6c**) as well as a heatmap depicting T cell marker expression to highlight T cells on the Umap (**Fig S6d**). Furthermore, we have analyzed whether the

cluster identified as T cells, is significantly more abundant in APP23 mouse brains compared to controls. This analysis has been incorporated into Fig S6, and presented in a new figure panel (**Fig S6e**), where we clearly see an increase in abundance in APP23-late mice compared to other groups.

Fig S6b:

Fig S6c:

Fig S6d:

Fig S6e:

8)

In the context of CXCL10-CXCR3 mediated T-cell chemotaxis, our data show that:

1. Interferon-stimulated (ISG⁺) CD8⁺ T cells produce high levels of **CXCL10 (Fig 4b-c)** which is a ligand for CXCR3
2. **CXCR3** is robustly expressed on infiltrating effector T cells at late disease stages (**Fig 4b-c**)
3. and previous studies have reported **CXCR3⁺ T cells** infiltrating Alzheimer's disease (AD) brains (Gate et al., *Nature*, 2020).

In addition, recent work has further implicated CXCL10-CXCR3 interactions in shaping CNS immune infiltration patterns in neurodegeneration (Zhou et al., *Nat Neurosci*, 2025). Together with established literature describing CXCL10 as a key ligand for CXCR3-mediated chemotaxis (Groom & Luster, *Nat Rev Immunol*, 2011), these findings led us to **hypothesize** a role for CXCL10-CXCR3 signaling in shaping T-cell infiltration patterns in neuroinflammation.

Our data strongly indicate that the CXCR3-CXCL10 axis contributes to T-cell infiltration during neuroinflammation, although we do not claim definitive causality at this stage. In response to the reviewer's feedback, we have revised the relevant statements in the manuscript to clarify that this is a **proposed mechanism** supported by correlative evidence. We agree that functional studies would be required for a conclusive mechanistic demonstration. To further support our proposed mechanism, we have now included additional experimental evidence that supports this interpretation. We performed functional transwell migration assays to assess the

We performed functional transwell migration assays, in which splenic T cells from Cas9-expressing mice were retrovirally transduced with CXCR3-targeting or non-targeting (NT) sgRNAs (marked by BFP), achieving ~40% transduction and a clear reduction of CXCR3 surface expression in CXCR3 KO cells. When CXCL10 was

added in the basolateral chamber, we observed strong migration of NT cells, whereas migration was significantly reduced in CXCR3 KO cells. Neutralizing CXCR3 in NT cells similarly decreased migration, confirming CXCR3-dependent responsiveness to CXCL10 in the context of T cell infiltration.

We next tested whether ISG-induced CD8⁺ T cells themselves act as chemotactic stimuli. CD8⁺ T cells in the basolateral chamber were treated with DMXAA for 5 hours to induce a STING-dependent ISG response (Ishikawa & Barber, Nature 2008; Larkin et al., J Immunol. 2017), and CXCR3 KO or NT cells were seeded into the insert. ISG-induced T cells elicited robust transmigration, which was markedly reduced in CXCR3 KO cells.

These functional data demonstrate that ISG-induced CD8⁺ T cells possess chemotactic activity that is at least partially dependent on CXCR3. While not excluding additional pathways, our results indicate that the CXCL10–CXCR3 axis represents a relevant and mechanistically significant contributor to T-cell infiltration in neuroinflammation.

Fig 4b-c:

statement in manuscript: *“Taken together, we identify ISG CD8 as a major source of CXCL10 in late amyloid disease and suggest that T-cell infiltration during neuroinflammation is driven by activation of CXCR3 by CXCL10.....”*

Fig 4h-i:

h

i

Figure legend:**h**) Shown are transwell migration assays comparing CXCR3 knockout (CXCR3 KO) and non-targeting control (NT KO) CD8⁺ T cells. The schematic on the left depicts the migration setup. The bar graph on the right shows the number of migrated cells (cells/ μ l) under different stimulation conditions (CXCL10, medium, with or without CXCL10 blocking antibody). **(i)** Quantification of migrated CXCR3 knockout (CXCR3 KO) and non-targeting control (NT KO) CD8⁺ T cells in transwell assays with basolateral compartments containing either untreated (UT) or ISG-induced CD8⁺ T cell supernatants. Data are presented as mean \pm SEM; each dot represents one biological replicate. For **(h)** and **(i)** statistical significance was assessed by two-way ANOVA with Tukey's multiple comparisons test (* $p < 0.0322$; ** $p < 0.0021$; *** $p < 0.0002$; **** $p < 0.0001$; ns = not significant)....

Fig S9a:

Figure legend: (a) Representative flow cytometry gating strategy for identifying BFP⁺ transduced T cells. Sequential gating was applied for lymphocytes, singlets, live cells, GFP⁺ and BFP⁺ populations. The box highlights the percentage of BFP⁺ cells, indicating successful transduction. **(b)** Flow cytometric analysis of CXCR3 surface expression in T cells transduced with three different CXCR3-targeting gRNAs (g1–g3), a non-targeting (NT) control guide, or non-transduced. CXCR3 (APC, y-axis) versus eBFP (x-axis) staining shows reduced CXCR3 signal in CXCR3 gRNA-transduced cells compared to controls. **(c)** Histograms showing CXCR3 expression in GFP⁺ T cells transduced with CXCR3 gRNAs (green), NT guide (gray), or non-transduced cells (orange). Left: CXCR3 expression in all GFP⁺ T cells. Right: comparison of CXCR3 levels between BFP⁺ (successfully transduced) and BFP⁻ (non-transduced) subsets for CXCR3 g1 and NT guide conditions, confirming efficient knockout in BFP⁺ cells. **(d)** Quantification of migrated T cells in response to CXCL10 in a transwell assay. CXCR3 knockout (green bars) significantly reduced migration toward CXCL10 compared to NT controls (gray bars). Migration was abrogated by CXCR3 blocking antibody (“Block”) and not observed in medium-only controls. Data represent mean ± SD; statistical significance was assessed by two-way ANOVA with Tukey’s multiple comparisons test, ****p < 0.0001, ns = not significant.

9)

Our conclusion was based on converging evidence from PD1 expression patterns in the human spatial transcriptomics data and corresponding observations in the mouse model. We agree that while these findings strongly suggest an activation-induced exhaustion trajectory, definitive proof of this fate requires further functional characterization. Accordingly, we have refined our wording to more accurately reflect this inference, as follows:

“As *PDCD1* is mostly expressed by exhausted T cells, we hypothesize that similar to our pre-clinical observation, T cells verging on plaques in human CNS amyloid pathology putatively undergo activation-induced exhaustion.....”

The central aim of this study was to delineate the temporal dynamics of ISG CD8 T cells during progressive amyloidosis and their contribution to the evolving immune microenvironment. We agree that a deeper investigation into the transcriptional fate and phenotypic modulation of infiltrating T cells in response to local cues will be highly informative, and we are actively pursuing this in ongoing work.

Minor comments:

We have now corrected for the missing information as follows:

Fig S4a: We have now added labels clarifying the different conditions plotted on the x-axis.

Fig S4a:

Fig 3f: The missing labels for the four distinct groups on the right-side dot plot have been added.

Fig 3f:

Fig 6c: P1, P2, P3, and P4 refer to "distinct patients". We have now clarified this in the figure and the figure legend to ensure better interpretability.

Figure legend: "Data was scaled and plotted for each patient separately. P = Patient."

Fig 6c:

Response to Reviewer #2

1)

To address the contribution of T cell subtypes, we have employed a combined single-cell and spatial transcriptomics approach in APP23 mice, which allows for both transcriptional and spatial resolution of T-cell subsets. From our single-cell RNA-seq data, we show that the interferon-stimulated gene (ISG) signature observed in brain-infiltrating T cells is almost exclusively derived from CD8⁺ T cells (see **Fig 1b**, **Fig S2c**, **Fig 3f**). This indicates that CD8⁺ T cells are the dominant ISG-expressing population in the diseased brain. We also describe the IL17-producing CD4⁺ T cell population. To benchmark the proportion of CD4 T cells more clearly in our model, we have now aligned the scale of T cell proportions in **Fig 1c**. From the very low proportion of CD4 T cells in our model we have concluded that CD4 IHC will be largely negative and the transcriptional single cell seq analysis is much more informative guiding towards CD8+ CD3+ T cells.

Fig 1b:

Fig S2c:

Fig 3f:

Fig 3g-i:

2)

In addition to our transcriptomic and spatial sequencing analysis, we now provide direct experimental evidence supporting that ISG⁺ CD8⁺ T cells can promote CXCR3-dependent T-cell chemotaxis.

We performed functional transwell migration assays. Splenic T cells from Cas9-expressing mice were retrovirally transduced with CXCR3-targeting or non-targeting (NT) guide RNAs (marked by BFP), achieving ~40% transduction and a clear reduction of CXCR3 surface expression in CXCR3 KO cells. CXCL10 in the basolateral chamber induced strong migration of NT cells, whereas migration was significantly reduced in CXCR3 KO cells. Neutralizing CXCR3 in NT cells similarly decreased migration, confirming CXCR3-dependent responsiveness to CXCL10.

We next tested whether ISG-induced CD8⁺ T cells themselves act as chemotactic stimuli. CD8⁺ T cells in the basolateral chamber were treated with DMXAA for 5 hours to induce a STING-dependent ISG response (Ishikawa & Barber, Nature 2008; Larkin et al., J Immunol. 2017), and CXCR3 KO or NT cells were seeded into the insert. ISG-induced T cells elicited robust transmigration, which was markedly reduced in CXCR3 KO cells.

These functional data demonstrate that ISG-induced CD8⁺ T cells possess chemotactic activity that is at least partially dependent on CXCR3. While not excluding additional pathways, our results indicate that the CXCL10–CXCR3 axis represents a relevant and mechanistically significant contributor to T-cell infiltration in neuroinflammation.

Regarding the suggestion to use a CXCL10-blocking antibody to disrupt the CXCL10–CXCR3 interaction, we instead employed a CXCR3-blocking antibody (Fig. 4h, Fig. S9d) to achieve the same objective. As shown, addition of the blocking antibody to the apical compartment (insert) markedly reduced the infiltration of NT KO T cells in response to CXCL10, demonstrating that blocking CXCR3 effectively attenuates the chemotactic response to CXCL10.

Regarding the suggested genetic mouse models (e.g., CD8-cre;Cxcl10;APP23), we fully acknowledge the value of such in vivo approaches. However, generating and aging these mice to the relevant 12-24 month time points required for neurodegeneration studies in APP23 is not feasible within the timeframe of this revision. Nevertheless, we are currently breeding appropriate genetic models to pursue this question in a dedicated follow-up study.

Taken together, the new transwell migration assay data, our spatial transcriptomics results, and prior literature all converge to support the role of CXCL10-CXCR3 signaling in orchestrating CD8⁺ T cell infiltration during neuroinflammation in APP23 mice.

Fig 4h-i:

figure legend: ...h) Shown are transwell migration assays comparing CXCR3 knockout (CXCR3 KO) and non-targeting control (NT KO) CD8⁺ T cells. The schematic on the left depicts the migration setup. The bar graph on the right shows the number of migrated cells (cells/ μ l) under different stimulation conditions (CXCL10, medium, with or without CXCL10 blocking antibody). *(i)* Quantification of migrated CXCR3 knockout (CXCR3 KO) and non-targeting control (NT KO) CD8⁺ T cells in transwell assays with basolateral compartments containing either untreated (UT) or ISG-induced CD8⁺ T cell supernatants. Data are presented as mean \pm SEM; each dot represents one biological replicate. For *(h)* and *(i)* statistical significance was assessed by two-way ANOVA with Tukey's multiple comparisons test (* $p < 0.0322$; ** $p < 0.0021$; *** $p < 0.0002$; **** $p < 0.0001$; ns = not significant).

Fig S9a:

figure legend: **(a)** Representative flow cytometry gating strategy for identifying BFP⁺ transduced T cells. Sequential gating was applied for lymphocytes, singlets, live cells, GFP⁺ and BFP⁺ populations. The box highlights the percentage of BFP⁺ cells, indicating successful transduction. **(b)** Flow cytometric analysis of CXCR3 surface expression in T cells transduced with three different CXCR3-targeting gRNAs (g1–g3), a non-targeting (NT) control guide, or non-transduced. CXCR3 (APC, y-axis) versus eBFP (x-axis) staining shows reduced CXCR3 signal in CXCR3 gRNA-transduced cells compared to controls. **(c)** Histograms showing CXCR3 expression in GFP⁺ T cells transduced with CXCR3 gRNAs (green), NT guide (gray), or non-transduced cells (orange). Left: CXCR3 expression in all GFP⁺ T cells. Right: comparison of CXCR3 levels between BFP⁺ (successfully transduced) and BFP⁻ (non-transduced) subsets for CXCR3 g1 and NT guide conditions, confirming efficient knockout in BFP⁺ cells. **(d)** Quantification of migrated T cells in response to CXCL10 in a transwell assay. CXCR3 knockout (green bars) significantly reduced migration toward CXCL10 compared to NT controls (gray bars). Migration was abrogated by CXCR3 blocking antibody (“Block”) and not observed in medium-only controls. Data represent mean ± SD; statistical significance was assessed by two-way ANOVA with Tukey’s multiple comparisons test, ****p < 0.0001, ns = not significant.

3)

We would like to clarify that we feel much of the requested information is already included in the current version of the manuscript. We explain below in detail how each point has been addressed and what additional changes we have incorporated to improve clarity.

TCR clonal distribution: In **Fig 5e** we show the distribution of TCR-clones based on their size between different stages and groups for each of the relevant CD8 T cell clusters. The color legend indicates the size of the clones, meaning the number of identified T cells that have the same TCR and belong to one clone. The y-axis denotes the number of such clones within each size-range for each of the analyzed groups. Taken together, the data shows greater TCR clonality in late disease, with a higher proportion of expanded clones contributing to the overall repertoire.

We have updated the corresponding legend in **Fig 5e** to improve clarity of the associated TCR clonality analysis.

Fig 5e:

Figure legend: “**(e)** Number of T-cell clones grouped by clone size (visualized by color gradient) in identified T cell subsets between all cohorts.”

Top clones in proliferating T cells: While we do not have a distinct cluster labeled as "proliferating T cells," our pathway analysis (IPA) revealed upregulation of proliferation-associated markers in T cells. In addition, expansion of TCR clones - large size of clones - is indicative of T cell proliferation. The distribution of such expanded clones between groups for each Cd8 cluster is shown in **Fig 5e**, where Cd8_1 have the highest proportion of expanded clones (see **Fig 5e** above).

For reference, we have now included UMAP plots showing the distribution of the top 10 clones in APP23-late among the clusters in a new supplementary - **Fig S8**.

Fig S8:

Additionally, to avoid any confusion while interpreting this data, we have edited the language in the Results section when referring to **Fig 5e**. We are now confident that with the new modifications, we enhance the accessibility of our findings to a broader audience.

edited text: "In late disease, we observed markedly increased TCR clonality, with a higher proportion of expanded clones contributing to the overall repertoire. Notably,

the Cd8_1 cluster contained many highly expanded clones (>200 cells) with the highest frequency of these expanded clones observed in the APP23-tg late group, as illustrated in Fig 5e.”

Peptide prediction: We appreciate the reviewer’s interest in the functional specificity of the expanded TCRs. However, reliable antigen prediction from TCR sequences in mice remains an open and technically challenging area and beyond the current ‘state-of-art’. Peptide-TCR binding datasets are limited for murine MHC molecules and the few prediction tools that exist are primarily trained on human repertoires. For this reason, we believe peptide prediction is outside the scope of this study and would not provide robust or biologically meaningful insights at this stage.

4)

Our data do indeed show an increased presence of IL17+ T cells in late APP23 mice. However, the primary focus of the current study was on interferon-stimulated gene (ISG)-expressing CD8+ T cells, which were found to be a key immunological feature in disease progression (See also comment 1).

While the IL17+ population appears to increase with disease progression, their etiology and impact remain open questions. We chose not to explore this for the current manuscript, as it fell beyond the scope of our study. That said, we fully agree that their potential role in neuroinflammation and neuronal dysfunction is of great interest. This is currently the focus of a follow-up investigation in our lab aimed at elucidating how IL17+ T cells may contribute to neuroimmune interactions and neuronal health in this model. We anticipate that these future studies will provide a more comprehensive understanding of the function of T cells in different transcriptional fates and their relevance in amyloid beta pathology.

We have, however, included differential gene expression analysis of IL17+ cells and NKT cells between different stages as reference in the rebuttal to enable the reviewer to make additional inferences which may be of interest to them:

IL17+ T cells:
APP23 late vs APP23 early

IL17a+ cells APP23 late vs. APP23 early

EnhancedVolcano

● NS ● Log₂ FC ● p-value ● p-value and log₂ FC

total = 10466 variables

APP23 late vs WT late

IL17a+ cells APP23 late vs. Wt late

EnhancedVolcano

● NS ● Log₂ FC ● p-value ● p-value and log₂ FC

total = 9624 variables

NKT cells:
APP23 late vs APP23 early

NKT APP23 late vs. APP23 early

EnhancedVolcano

● NS ● Log₂ FC ● p-value ● p-value and log₂ FC

APP23 late vs WT late

NKT APP23 late vs. Wt late

EnhancedVolcano

● NS ● Log₂ FC ● p-value ● p-value and log₂ FC

Response to Reviewer #3

We thank the referee for acknowledging our high end bioinformatic analysis. We now provide additional experimental data validating the novel finding of an ISG-T intercellular mechanism that leads to dynamic T cell recruitment in amyloid pathology in murine and human systems. We think that our study now strongly goes beyond description (see new **Fig 4h,i** and **Fig S9** and also below in more detailed comments). We acknowledge that the reference the reviewer provided (Altendorfer et al., J Immunol. 2022 Oct 1;209(7):1272-1285. doi: 10.4049/jimmunol.2100737) was the first to identify Type-I interferon expression on T cells in the context of AD mouse models and we have corrected our citation to adequately reflect that.

Brain CD8⁺ T cells in APP-PS1 and aged WT mice showed similar differentially regulated genes as brain Trm CD8⁺ T cells in mouse models of acute virus infection, chronic parasite infection, and tumor growth, suggesting a common adaptive immune response in various inflammatory conditions of the CNS.

The novelty of our work in comparison, lies in the spatio-temporal resolution which identifies ISG CD8⁺ T cells as a late disease phenomenon which exclusively localize around Amyloid plaques. Our observations that the ISG response shifts from microglial in early disease to T-cell based in late disease adds another important layer in understanding how Type-I interferon regulated inflammation develops with disease progression. We additionally identify ISG CD8⁺ T cells as the source of CXCL10 which is a potent T-cell chemoattractant. These findings are not just a corroboration of Altendorfer et. al, but expand the knowledge on ISG CD8⁺ T cells in how they modulate immune responses in Abeta pathologies, which was previously not known. This, combined with the observation that the localization of ISG CD8⁺ T cells correlates with the influx and localization of other effector CD8⁺ T cells, has significant implications in the targetability and the consequent approach in targeting Type-I interferon related inflammation in amyloid disease.

While a major goal of our study was to generate a high-resolution, spatio-temporal map of the CD8⁺ T-cell immune response during amyloid disease development, particularly the emergence of ISG⁺ CD8⁺ T cells, we fully agree that descriptive analyses alone cannot establish mechanistic causality.

Our approach intentionally leveraged unbiased spatial transcriptomics and integrative bioinformatic analyses to infer putative mechanisms without relying solely on predefined cell-type annotations, which can be confounded by expression-level variability. Using these methods, we were able to localize ISG transcripts with high spatial specificity around plaques at late disease stages and identify ISG⁺ CD8⁺ T cells as the predominant CXCL10-expressing population. This spatial correlation, combined with the co-occurrence of CXCR3⁺ effector CD8⁺ T cells, led us to hypothesize that the CXCL10-CXCR3 axis contributes to T-cell infiltration in the amyloidogenic brain.

To address the reviewer's concern directly, we now complement these descriptive findings with new functional evidence, demonstrating that this axis is indeed capable of mediating T-cell chemotaxis:

CXCL10-CXCR3-dependent T-cell migration.

We performed transwell migration assays using splenic T cells from Cas9-expressing mice, retrovirally transduced with CXCR3-targeting or non-targeting (NT) guide RNAs (BFP⁺). CXCL10 in the basolateral chamber induced strong migration of NT cells, whereas migration was significantly reduced in CXCR3-deficient cells. Pharmacologic CXCR3 blockade in NT cells similarly reduced migration, demonstrating that CXCL10-dependent chemotaxis requires CXCR3.

ISG-induced CD8⁺ T cells act as chemotactic stimuli.

To test whether ISG⁺ CD8⁺ T cells themselves promote chemotaxis, we stimulated CD8⁺ T cells with DMXAA to induce a STING-dependent ISG response (Ishikawa & Barber, Nature 2008; Larkin et al., J Immunol. 2017) and placed them in the basolateral chamber. These ISG-induced T cells triggered robust transmigration of NT responder cells, which was markedly diminished when responder cells lacked CXCR3. This shows that ISG⁺ CD8⁺ T cells create a chemotactic milieu that is at least partially CXCR3-dependent.

Together, these *in vitro* experiments substantiate the mechanism we inferred from our spatial and transcriptional analyses: ISG⁺ CD8⁺ T cells are capable of contributing to CXCR3-dependent T-cell recruitment.

At the same time, we recognize that establishing causality *in vivo*, such as through CXCL10-neutralizing approaches or genetic models like CD8-Cre;Cxcl10;APP23, would constitute the next step. As discussed in our revised manuscript, generating and aging such multi-allelic lines to the required 12-24-month timepoint is not feasible within this revision cycle, but we are actively pursuing these models in ongoing work.

In summary, while the descriptive framework of our study provided the foundation for hypothesis generation, the newly added functional assays now offer direct mechanistic support for the proposed role of the CXCL10-CXCR3 axis in facilitating T-cell infiltration during amyloid pathology.

Fig 4h-i:

h

i

figure legend: **h**) Shown are transwell migration assays comparing CXCR3 knockout (CXCR3 KO) and non-targeting control (NT KO) CD8⁺ T cells. The schematic on the left depicts the migration setup. The bar graph on the right shows the number of migrated cells (cells/ μ l) under different stimulation conditions (CXCL10, medium, with or without CXCL10 blocking antibody). **(i)** Quantification of migrated CXCR3 knockout (CXCR3 KO) and non-targeting control (NT KO) CD8⁺ T cells in transwell assays with basolateral compartments containing either untreated (UT) or ISG-induced CD8⁺ T cell supernatants. Data are presented as mean \pm SEM; each dot represents one biological replicate. For **(h)** and **(i)** statistical significance was assessed by two-way ANOVA with Tukey's multiple comparisons test (* $p < 0.0322$; ** $p < 0.0021$; *** $p < 0.0002$; **** $p < 0.0001$; ns = not significant).

Fig S9a:

figure legend: **(a)** Representative flow cytometry gating strategy for identifying BFP⁺ transduced T cells. Sequential gating was applied for lymphocytes, singlets, live cells, GFP⁺ and BFP⁺ populations. The box highlights the percentage of BFP⁺ cells, indicating successful transduction. **(b)** Flow cytometric analysis of CXCR3 surface expression in T cells transduced with three different CXCR3-targeting gRNAs (g1–g3), a non-targeting (NT) control guide, or non-transduced. CXCR3 (APC, y-axis) versus eBFP (x-axis) staining shows reduced CXCR3 signal in CXCR3 gRNA-transduced cells compared to controls. **(c)** Histograms showing CXCR3 expression in GFP⁺ T cells transduced with CXCR3 gRNAs (green), NT guide (gray), or non-transduced cells (orange). Left: CXCR3 expression in all GFP⁺ T cells. Right: comparison of CXCR3 levels between BFP⁺ (successfully transduced) and BFP⁻ (non-transduced) subsets for CXCR3 g1 and NT guide conditions, confirming efficient knockout in BFP⁺ cells. **(d)** Quantification of migrated T cells in response to CXCL10 in a transwell assay. CXCR3 knockout (green bars) significantly reduced migration toward CXCL10 compared to NT controls (gray bars). Migration was abrogated by CXCR3 blocking antibody (“Block”) and not observed in medium-only controls. Data represent mean ± SD; statistical significance was assessed by two-way ANOVA with Tukey’s multiple comparisons test, ****p < 0.0001, ns = not significant.

Response to Reviewer #4

1)

The anatomical region shown in **Fig 2a** is the cerebral cortex of APP23 transgenic mice, which we have now specified in the revised figure legend. Regarding the apparent low cell density in the DAPI channel, this difference is due to the distinct tissue processing methods used for each figure. **Fig 2a** utilizes formalin-fixed, paraffin-embedded (FFPE) tissue sections, which typically exhibit lower cell density due to tissue shrinkage and cell loss during processing. In contrast, **Fig 2e** displays images from fixed-frozen tissue sections, a method that generally preserves tissue architecture and cell density more effectively. Since this crucial distinction was already present in the legends, we'll emphasize how these different processing methods inherently lead to the observed differences in cell density and overall appearance. We've ensured that the method used for tissue section preparation for each of the image panels is prominent in the revised figure legend for clarity.

Fig 2a:

Fig 2e:

Figure legend of Fig 2a: “(a) Shown are zoomed-in areas of interest of fresh frozen paraffin-embedded (FFPE) cerebral cortex slides from APP23-tg mice aged 8, 12, 16, 20, and 24 months.....”

Figure legend of Fig 2e: “(e) Fixed frozen immunofluorescence (IF) brain cortex sections of one exemplary 24-month-old APP23-tg (upper panel) and one wildtype (lower panel) mouse.”

2)

The reviewer is correct in noting the high proportion of T cells depicted in **Fig 2f**. This question is in line to the one raised by Reviewer 1 (comment 6).

As we explained above, this figure represents a plaque-dense region within the cerebral cortex, and our analysis indeed shows that T cells are a predominant cell type in these specific areas. To address the reviewer's point about the generality of this observation, we have now extended our analysis to include a quantification of T cells in plaque-devoid regions within the cerebral cortex (**Fig S5c**). Our new data proves that T cells are sparse in these plaque-devoid regions, which we had already visually represented in the lower panel of **Fig 2f**. Therefore, the image in the upper panel of **Fig 2f** is an exemplary illustration of a plaque-dense region where T cells are highly concentrated and do indeed dominate. Regarding the quantification of T cell density across different anatomical regions, our study specifically focused on the cerebral cortex because it's the primary area where amyloid plaques form in our model. Our goal was to investigate the cellular environment immediately surrounding these plaques, making the cerebral cortex the most relevant region for our analysis. We have now quantified and presented data comparing T cell density in both plaque-dense and plaque-devoid areas within the cerebral cortex to provide a more complete and accurate representation of T cell distribution in this key region.

Fig 2f:

Fig S5c:

3)

We thank the reviewer for pointing out this discrepancy. This comment raised by the reviewer was the same as by Reviewer 1, comment 6, we have addressed it in detail above.

First, we'd like to address the specific point about **Fig 1c**. The reviewer correctly identified a labeling error; the original percentages of CD3+ T cells within the CD45+ population (e.g., 0.1%, 0.2%) were indeed incorrect and should have been higher by a factor of 100 (e.g., 10%, 20%). We've corrected this in the revised **Fig 1c** to accurately reflect these percentages.

Fig 1c:

Regarding the apparent overestimation of T cells in our initial immunohistochemical analysis (**Fig 2b**) compared to the scRNA-seq data (**Fig 1c**), this occurred because our initial analysis of immunohistochemical stainings focused on A β plaque-dense regions to visualize and quantify A β plaque-associated T cells. Since these specific regions are highly abundant in T cells, this indeed led to an overestimation of T cells relative to total cells when generalized. To compensate for this, we now re-analyzed plaque-devoid regions in sections from the same animals and show that these regions are also scarce in T cells (**Fig S5c**). Taken together, the quantification of CD3+ T cells evens-out relative to all cells in the immunohistochemical stainings and resolves the discrepancy between the scRNAseq data and immunohistochemistry.

Fig 2b:

Fig 1c:

Fig S5c:

4)

Our original phrasing implied a mechanistic “attraction”, whereas the data more accurately reflect a stronger correlative association between T cells and parenchymal plaques relative to vascular amyloid deposits. The reviewer’s quantitative restatement (showing that for every 10 vascular deposits T cells/total cells \approx 0.1-0.15, compared with \approx 0.05 for 10 parenchymal plaques) captures this relationship well. To ensure our language aligns precisely with what the data support, we have revised the manuscript accordingly. We have removed the term “attraction” and now describe the finding as a preferential association of T cells with parenchymal plaques. This wording accurately conveys the correlative nature of the observation without implying an untested mechanistic process.

We believe this refinement strengthens the clarity and rigor of our interpretation of **Fig 2g**.

Fig 2g:

g

Revised section: “*The number of T-cells quantified per histological section significantly correlated with plaque burden, with a preferential association with parenchymal plaques compared to vascular amyloid deposits (Fig 2g)*”

5)

We have added additional methodical details on how the Molecular Cartography technique was used in our study. We have now included a citation for the Molecular Cartography technique (Groiss et al., 2021, doi: [10.1101/2021.10.11.463936](https://doi.org/10.1101/2021.10.11.463936)) and expanded the methodology section to describe its principles more clearly. This method has been effectively used in multiple high-impact studies. (Guilliams et al., 2022, Cell doi: [10.1016/j.cell.2021.12.018](https://doi.org/10.1016/j.cell.2021.12.018); Legnini et al., 2023, Nature Methods doi: [10.1038/s41592-023-01986-w](https://doi.org/10.1038/s41592-023-01986-w); Bravo Gonzalez-Bras et al., 2023, Nature Methods doi: [10.1038/s41592-023-01938-4](https://doi.org/10.1038/s41592-023-01938-4)).

Specifically, we now explain that Molecular Cartography is an imaging-based, highly multiplexed single-molecule fluorescence in situ hybridization (smFISH) technology designed to detect and quantify individual RNA transcripts with subcellular resolution while preserving tissue morphology. The technique works in principle by employing a proprietary system of combinatorial barcoded probes and sequential imaging rounds. Briefly, custom designed, gene-specific oligonucleotide probes, each bearing a unique barcode, are hybridized to target RNA molecules within tissue sections. In subsequent iterative imaging cycles, these barcodes are decoded through the sequential binding and unbinding of fluorescent reporter molecules. Each cycle involves the detection of specific fluorophores associated with different barcode bits. By recording the fluorescent signal across multiple rounds, a unique "fingerprint" is generated for each

RNA molecule, allowing its identification and precise spatial localization. This imaging-based method achieves subcellular resolution (approximately 300 nm), enabling the visualization of individual transcripts and their spatial context within cells and tissues. Regarding the error rate of the Molecular cartography platform, according to the company, the platform achieves an average specificity between 99.45% and 99.9%, underscoring the low error rate and high reliability of the technique (Groiss et al., 2021, doi: doi.org/10.1101/2021.10.11.463936).

For our experiments, tissue sections were prepared according to Resolve Biosciences' guidelines and processed in their service laboratory. Following data acquisition, transcripts were segmented to individual cells based on DAPI nuclear staining and Resolve Biosciences' proprietary cell segmentation algorithm. The distribution of transcripts per segmented cell was then analyzed using our custom bioinformatic pipelines.

This information has been incorporated into the revised manuscript in Methods section 'Targeted spatial transcriptomics. Sub-section: Methodology of Molecular Cartography platform'. We have also added a new illustration showing how ROIs were selected for Molecular Cartography (revised **Fig S6a**) to improve clarity. Additional general workflow graphics are also available at the Platform's website (<https://www.resolvebiosciences.com/>).

Fig S6a:

a

6)

Distance 0 is determined as the distance to the boundary of the closest plaque. This is done by calculating a distance map based on the segmentation of all plaques within one image. This is then used to capture the smallest distance of each transcript to the closest plaque. All transcripts were treated the same and assigned the distances based on the same distance maps.

Fig 3g:

g

Fig 3i:

7)

The analyzed anatomical region in **Fig 3** and **Fig 4** was the cerebral cortex in line with the analyses shown in **Fig 2**. The image in **Fig 3g** is a superimposed image generated from Molecular Cartography output. It combines the raw transcript signal, DAPI staining (nuclei), pFTAA staining (amyloid plaques), and cell segmentation boundaries (in white), to allow spatial localization of gene expression relative to individual cells and plaques. All layers were co-registered and overlaid using ImageJ (Fiji), with brightness and contrast adjusted uniformly for visualization. No further image processing or filtering was applied beyond overlay and visualization steps.

Consistency of A β detection: In **Fig 2**, amyloid plaques were detected using an anti-A β antibody, whereas in the Molecular Cartography assay (**Fig 3g**), plaques were labeled with pFTAA, a fluorescent tracer that binds β -sheet-rich aggregates. The selection of pFTAA for Molecular Cartography was based on its compatibility with multiplexed RNA detection protocols, as it allows non-immunogenic, label-free staining of dense-core amyloid plaques without interfering with transcript signal detection. In contrast, antibody-based detection is optimal for classical immunohistochemistry but not compatible with the tissue handling and fluorescence chemistry of spatial transcriptomics.

Fig 3g:
g

Differentiation of parenchymal plaques and vascular amyloid deposits: Indeed, we were able to visually see vascular amyloid deposits based on morphology, however, through spatial transcriptomics, our aim was to analyze T-cell phenotype describing transcript abundance and how they spatially distribute in association with A β overall. Therefore, our analysis combines all amyloid deposits. We did not differentiate between the two for 2 reasons:

1. The number of vascular amyloid deposits are fewer than parenchymal plaques as shown in **Fig 2c**. The size of the sections with respect to the size of the ROIs that can be analyzed for the spatial transcriptomics are not as large as tile scans used for immunofluorescence stainings as in **Fig 2**. Therefore, the number of vascular amyloid deposits that would be quantifiable would be low, making it difficult to draw any conclusive interpretations on additional niche-specific differences.

Fig 2c:

2. Based on our single cell data, our focus was to understand how T-cell response is conditioned towards Abeta in general in a spatial context. This would serve as first line evidence that transcriptional programmes are altered in T cells on Abeta exposure. Niche specific differences between vascular and parenchymal niches is definitely an interesting scientific question and part of a follow-up study that requires experimental setups which were not within the scope of the framework presented here.

8)

We confirm that we performed T cell subclustering within the spatial transcriptomics data, and the results are now shown in **Fig. S6f**. This analysis identified multiple transcriptionally distinct T-cell subsets, including ISG⁺ CD8⁺ T cells, which spatially colocalize with amyloid plaques in late disease stages. These findings further support the relevance of our observations and highlight the ability of spatial transcriptomics to resolve immune cell states *in situ* (**Fig S6h**).

In response to the reviewer's concerns about global clustering and neuronal representation in **Fig S6c**, we have now revisited and refined the global clustering analysis. The updated panel includes an improved clustering scheme and an accompanying dot plot (**new Fig S6b–c**) showing the expression of canonical marker

9)

Clustering and cell annotations for the human spatial transcriptomics data, analogous to **Fig 3-4** and **Fig S6**, was not feasible due to the inherent characteristics of post-mortem human tissue

The RNA integrity in such samples typically results in a limited number of detectable transcripts per cell, which constrains resolution for high-dimensional analyses. As a result, several key markers did not reach the expression thresholds required for reliable cell-wise clustering or dimensionality reduction (e.g., UMAP). Under these conditions, unsupervised clustering or cell-type annotation would not yield robust or interpretable results. Therefore, we focused our analysis on transcripts with consistent detection and on quantifying their spatial relationships to amyloid plaques, which provided the most reliable and biologically meaningful insights from the available material.

10)

Thank you for pointing out the "xx" in the IPA section of the Methods. We have corrected it to "Canonical Pathways".

Point-by-point response

“Type-I Interferon drives T-cell responses to Amyloid-beta in the central nervous system”

Reviewer #1 (Remarks to the Author)

The authors have adequately addressed most of my previous comments. I still have several questions related to my initial comment number 7 regarding the single-cell spatial transcriptomics analysis.

In Figure S6b, the authors annotate a relatively large cluster as T cells. In contrast, the proportions of T cells reported across groups in Figure S6e appear quite low: in Tg late the proportion is approximately 0.01, and in the other groups it is around 0.002–0.003. Visually, however, the cluster annotated as T cells in Figure S6b appears larger than 1% of all cells. The authors should verify whether the reported proportions are accurate and clarify any potential discrepancy between the UMAP visualization and the quantitative estimates.

The authors also state:

“Similarly, cell-segmentation-based analyses confirmed an ISG-upregulating T-cell subset. When compared to other cells, the abundance of cells within this subset was higher in the plaque neighborhood (<69 μ m) (Fig S6a-f).”

Given this claim, I think it would be important to show the expression of canonical T-cell markers within this ISG-upregulating subset. This would help confirm that these cells are indeed T cells rather than another ISG-high population.

Reviewer #3 (Remarks to the Author)

Congratulations to the authors. The revised version shows an immense improvement and an overwhelming amount of additional data now clearly supporting the authors' conclusion. All aspects and comments raised on the initial submission are adequately addressed.

Reviewer #4 (Remarks to the Author)

The author addressed most of my concerns.

As the authors propose a conceptual shift from microglia to T cells, and given that spatial transcriptomic data are available, it would be highly informative to leverage these data more fully. Specifically, the authors could examine age-dependent changes in the spatial distribution and density (or cellular composition) of different immune cell types as a function of distance from amyloid plaques. An analysis analogous to the stacked cell-composition plot in Figure 1, but incorporating spatial proximity to plaques, would be particularly compelling. Without such analyses, the spatial transcriptomic data are underutilized and do not substantially extend the insights already provided by the scRNA-seq results.

Response to Reviewer #1

It is correct that the frequencies originally shown in Fig. S6e were inconsistent with the relative size of the T-cell cluster visible in the UMAP in Fig. S6b. This discrepancy arose from an error in the normalization step used to calculate group-wise proportions. We have now corrected this analysis and updated Fig. S6e with the accurate frequencies, which are consistent with the UMAP visualization in Fig. S6b.

We now provide additional plots showing composite cell-type signature scores across all clusters. The ISG-upregulating subset is strongly enriched for T-cell markers, without enrichment for signatures corresponding to all other major cell types, including microglia, astrocytes, oligodendrocytes, or endothelial cells. These analyses further support the conclusion that the ISG-high population is of T-cell identity. Given the targeted nature of the spatial transcriptomics panel and variable detection of individual canonical markers across samples, we relied on composite signature scores derived from multiple markers to ensure robust and statistically reliable cell-type assignment.

Importantly, the primary spatial analyses presented in the main figures do not depend on cluster-based cell-type annotation of the spatial dataset. Instead, we applied a transcript-distance-based targeted approach to quantify spatial enrichment of predefined gene programs, consistent with our hypothesis-driven design and the structure of the targeted panel. The UMAP clustering shown in Fig. S6 was performed during revision in response to the reviewer's request and is therefore included as a complementary exploratory analysis rather than as the basis for our main conclusions.

Fig S6e:

Additional cell-type signature scores:

Response to Reviewer #3

We thank the reviewer for the positive and supportive assessment of our revised manuscript.

Response to Reviewer #4

In the original version of the manuscript, the spatial data were primarily used in a hypothesis-driven manner to test whether the T-cell phenotypes identified by scRNA-seq are spatially enriched in plaque neighborhoods, which directly addresses our central biological question. Within the scope of the specific aims of this study, we believe the spatial dataset was used in a manner aligned with this central biological question. At the same time, we fully agree with the reviewer that these datasets will be valuable for additional exploratory analyses, and we look forward to the community further leveraging them once they are publicly available. In this regard, we have now made the spatial transcriptomics dataset publicly available and have explicitly stated this in the Data Availability section of the revised manuscript.

In response to the reviewer's suggestion, we have now performed an additional analysis integrating cell-type composition with plaque distance across age groups, analogous to the stacked composition analysis in Fig. 1 but stratified by spatial proximity to plaques. These results are presented in the revised manuscript as new Supplementary **Fig. S6f** and **S6g**. As shown, this analysis is fully consistent with the model proposed in the manuscript and illustrates the same age-dependent shift in immune cell composition near plaques, from a predominantly microglia-associated response at earlier stages to increased T-cell representation at later stages.

Specifically, we added a paragraph in the *Results* section describing this: "...Importantly, this age-dependent shift was independently recapitulated in our targeted spatial transcriptomics dataset. Stratifying immune cell composition by proximity to A β plaques revealed a predominantly microglia-associated response in plaque-adjacent regions at early stages, which transitioned toward increased T-cell representation in late disease (Fig. S6f,g)...."

We believe that these additional analyses strengthen the spatial component of the study and further reinforce the conceptual shift proposed in the manuscript, while remaining consistent with the primary hypothesis-driven design of our spatial transcriptomics experiments.

Fig. S6f:

Figure legend: Stacked bar plots showing the proportional composition of major cell types identified by spatial transcriptomics stratified by distance from amyloid plaques (0-20 μm, 20-100 μm, and 100-500 μm) in APP23het mice at 13 months (n=2) and 24 months (n=2) of age.

Fig. S6g:

Figure legend: Line plots illustrating the proportion of microglia and T cells across increasing distances from amyloid plaques (0-20 μm, 20-100 μm, and 100-500 μm) in APP23het mice at 13 months (early, n=2) and 24 months (late, n=2).